# Activation of GPR3-β-arrestin2-PKM2 pathway in Kupffer cells stimulates glycolysis and inhibits obesity and liver pathogenesis

Ting Dong[1,2,6], Guangan Hu [1,6] ✉, Zhongqi Fan[1,3], Huirui Wang[2], Yinghui Gao[2], Sisi Wang[4], Hao Xu[4], Michael B. Yaffe [1], Matthew G. Vander Heiden [1,5], Guoyue Lv [3] ✉ & Jianzhu Chen [1] ✉

Kupffer cells are liver resident macrophages and play critical role in fatty liver disease, yet the underlying mechanisms remain unclear. Here, we show that activation of G-protein coupled receptor 3 (GPR3) in Kupffer cells stimulates glycolysis and protects mice from obesity and fatty liver disease. GPR3 activation induces a rapid increase in glycolysis via formation of complexes between β-arrestin2 and key glycolytic enzymes as well as sustained increase in glycolysis through transcription of glycolytic genes. In mice, GPR3 activation in Kupffer cells results in enhanced glycolysis, reduced inflammation and inhibition of high-fat diet induced obesity and liver pathogenesis. In human fatty liver biopsies, GPR3 activation increases expression of glycolytic genes and reduces expression of inflammatory genes in a population of disease-associated macrophages. These findings identify GPR3 activation as a pivotal mechanism for metabolic reprogramming of Kupffer cells and as a potential approach for treating fatty liver disease.

Non-alcoholic fatty liver disease (NAFLD), characterized by fat deposition in the liver, is the most common liver disorder globally. NAFLD progresses through a series of stages, from simple steatosis to non-alcoholic steatohepatitis (NASH) to cirrhosis[1,2]. Although the disease pathogenesis is not well understood, the development of NAFLD is highly correlated with obesity and diabetes and is pathogenically associated with lipid accumulation, inflammation, injury, and fibrosis in the liver. As NAFLD is also a metabolic disorder[3], mechanisms that link metabolism to inflammation could offer insights into the pathogenesis and help identify targets for therapeutic development[1].

Kupffer cells (KCs) are resident macrophages in the liver and the most abundant tissue-resident macrophages in the body. They play a key role in detoxification, pathogen removal, and tissue repair and homeostasis, but they can also contribute to the pathogenesis of liver diseases, including NAFLD, as they are involved in the initiation and progression of inflammation and tissue injury[4–6]. In response to local stimuli, KCs regulate both metabolic and immune functions and homeostasis in the liver[7]. Lipids and other metabolites have been shown to not only regulate the expression of genes associated with immune response in human macrophages[8,9] but also modulate the activation of KCs in models of fatty liver disease and steatohepatitis[10]. Recently, disease-associated macrophages (DAMs) have been identified by single-cell RNA sequencing (scRNAseq) in the livers of patients with advanced NAFLD (NASH and cirrhosis) and from mouse models of NASH[11–13]. DAMs exhibit altered expression of pathways not only associated with inflammation but also with metabolism, suggesting that metabolic reprogramming may be a promising strategy to treat NAFLD[14,15].

G protein-coupled receptors (GPCRs) play essential roles in metabolic disorders as they serve as receptors for metabolites and

[1]Koch Institute for Integrative Cancer Research and Department of Biology, Massachusetts Institute of Technology, Cambridge, MA 02139, USA. [2]Department of Natural Products Chemistry, School of Pharmaceutical Sciences, Shandong University, Jinan 250012, China. [3]Department of Hepatobiliary and Pancreatic Surgery, The First Hospital of Jilin University, Changchun 130021, China. [4]Department of Translational Medicine, The First Hospital of Jilin University, Changchun 130061, China. [5]Dana-Farber Cancer Institute, Boston, MA 02115, USA. [6]These authors contributed equally: Ting Dong, Guangan Hu ✉e-mail: gahu@mit.edu; lvgy@jlu.edu.cn; jchen@mit.edu

fatty acids[16]. In our screen for compounds that can reprogram macrophages, we found previously that diphenyleneiodonium (DPI), an agonist of GPR3, upregulates the expression of genes involved in glycolysis and lipid metabolism in macrophages[15]. GPR3 is broadly expressed in various tissues and has been shown to play important roles in metabolic processes[17–19]. Most recently, mice deficient in GPR3 were shown to develop late-onset obesity and GPR3 regulates thermogenesis in adipocytes[18,19], suggesting a role of GPR3 in regulating metabolism. GPR3 is considered a constitutively active orphan receptor that mediates sustained cAMP production in the absence of a ligand[20]. It has been shown that GPR3 stimulates the Aβ production by recruiting the scaffold protein β-arrestin2 (encoded by *Arrb2*) to regulate γ-secretase activity[21,22]. Interestingly, *Arrb2* expression is required for insulin sensitivity[23], and mice with *Arrb2* knockout in adipocytes are resistant to HFD-induced obesity[24], suggesting a role of β-arrestin2 in regulating metabolism. However, little is known about the function and mechanism of GPR3 signaling in regulating metabolism and in other cell types.

In this study, we have investigated the effect of GPR3 activation by DPI on the metabolic reprogramming of macrophages, the underlying molecular mechanisms, and the physiological effect on HFD-induced obesity and liver pathogenesis. We show: (i) DPI induces a rapid switch of cellular metabolism from oxidative phosphorylation (OxPhos) to glycolysis in macrophages by stimulating the formation of β-arrestin2-GAPDH-PKM2 super complex with greatly increased enzymatic activities; (ii) DPI also induces a prolonged increase in glycolytic activity by stimulating translocation of PKM2 from the cytosol to the nucleus, transactivation of c-Myc, and transcription of glycolytic genes; (iii) DPI inhibits HFD-induced obesity and liver pathogenesis in mice by stimulating glycolysis and suppressing inflammation in KCs, and in a manner that requires PKM2 expression in KCs; and iv) DPI also stimulates glycolysis and suppresses inflammation of KCs from patients with NAFLD. These findings identify the GPR3 to β-arrestin2 to PKM2 and to c-Myc pathway as a critical component of metabolic reprogramming in macrophages and activation of this pathway in KCs as a promising approach for treating obesity and NAFLD.

## Results

### DPI stimulates both rapid and sustained increase in glycolysis in macrophages

We have previously shown that DPI stimulates the transcription of many genes in the glycolysis pathway in human primary macrophages (Supplementary Fig. 1a, b)[15]. We confirmed the upregulation of hexokinase (HK), glyceraldehyde-3-phosphate dehydrogenase (GAPDH), lactate dehydrogenase A (LDHA) and enolase at the protein level in both human primary macrophages and an immortalized line of mouse Kupffer cells (ImKCs) in a DPI dose- and treatment time-dependent manner (Supplementary Fig. 1c). To investigate the effect of DPI on cellular metabolism, we measured cellular activities in glycolysis and OxPhos by assaying extracellular acidification rate (ECAR) and oxygen consumption rate (OCR), respectively, in ImKCs in the absence or the presence of 5, 50, and 500 nM DPI. Treatment of cells with DPI resulted in an immediate increase in ECAR and a concomitant decrease in OCR in a dose-dependent manner (Fig. 1a, b). (Because of the minimal effects of DPI at 5 nM, 50 and 500 nM DPI was used in all later experiments.) The DPI-stimulated increase in ECAR was sensitive to glucose, oligomycin, and 2-deoxylglucose (2-DG) and was associated with a significant increase in glycolytic capacity and reserve (Fig. 1c, d). Moreover, with or without extracellular glucose, DPI similarly stimulated ECAR (glycolysis), suggesting that DPI-induced immediate increase in glycolysis is not due to glucose uptake (Supplementary Fig. 1d). The effects of DPI on glycolysis and OxPhos was confirmed by quantifying the levels of the major intermediates in the glycolysis pathway and the tricarboxylic acid (TCA) cycle in ImKCs 6 h after DPI treatment. As shown in Fig. 1e, in a DPI dose-dependent manner,

intracellular glucose levels decreased significantly while the levels of intermediates in the glycolysis pathway, including glucose 6-phosphate (G6P), fructose 1,6-bisphosphare (F1,6BP), glyceraldehyde 3-phosphate (G3P), pyruvate, lactate, and acetyl-CoA increased significantly. In contrast, the levels of TCA cycle intermediates, including citrate, α-ketoglutarate (α-KG), succinate, fumarate, and malate, all decreased upon treatment with DPI in a dose-dependent manner. Similar changes in the levels of glucose, glycolysis, and TCA cycle intermediates were also seen 24 h after DPI treatment (Supplementary Fig. 1e). These results show that DPI dynamically regulates cellular metabolism of macrophages at two levels: rapid stimulation of glycolysis with concomitant inhibition of oxygen consumption and TCA cycle, and sustained increase in glycolysis by upregulating transcription and expression of genes in the glycolysis pathway.

### DPI stimulates glycolysis through GPR3 and β-arrestin2

DPI has been reported to be an agonist of GPR3 and an inhibitor of NADPH oxidases (NOX)[25,26]. We first examined whether DPI-stimulated glycolysis in macrophages required a fully functional NADPH oxidase. Bone marrow-derived macrophages (BMDMs) were prepared from *p47phox*[–/–] mice, which are defective in NOX activity since p47Pphox is the obligate organizer of the phagocyte NADPH oxidase (NOX2)[27]. Compared to wildtype (WT) BMDMs, *p47phox*[–/–] BMDMs had a significantly lower basal level of glycolysis, glycolytic capacity, and glycolytic reserve (Fig. 2a and Supplementary Fig. 2a, b). However, treatment with DPI increased the rate of glucose consumption and glucose uptake (Fig. 2b) in a dose-dependent manner and to a similar extent in both wildtype and *p47phox*[–/–] BMDMs. Similarly, DPI stimulated a similar increase in ECAR in ImKCs when NOX activity was pharmacologically inhibited by apocynin, a NOX-specific inhibitor (Fig. 2c). Together, these data suggest that DPI-stimulated glycolysis is independent of NOX activity.

To examine the requirement for GPR3 in DPI-stimulated glycolysis, BMDMs were prepared from *Gpr3*[–/–] mice. Compared to wild-type (WT) BMDMs, *Gpr3*[–/–] BMDMs had a significantly lower basal level of glycolysis, glycolytic capacity, and glycolytic reserve (Fig. 2d, e and Supplementary Fig. 2c, d). However, at 50 nM, DPI did not stimulate any increase in ECAR, glycolytic capacity, or glucose uptake in *Gpr3*[–/–] BMDMs as compared to WT controls. At 500 nM concentration, DPI stimulated a significant increase in glycolysis and glycolytic capacity in *Gpr3*[–/–] BMDMs, but the magnitude of increase was significantly lower than that seen in WT BMDMs, probably due to stimulation of other proteins by DPI at high concentration. Similarly, DPI at 50 nM did not stimulate any increase in ECAR, glycolytic capacity, and glucose uptake in GPR3-null ImKCs in which *Gpr3* was knocked out by CRISPR-Cas9 mediated gene editing or knocked down by siRNA (Supplementary Fig. 2e–j). Consistently, DPI (50 nM) failed to stimulate any significant increase in glucose consumption and glycolytic intermediates in *Gpr3*[–/–] ImKCs (Supplementary Fig. 2k). Moreover, sphingosine-1-phosphate (S1P) a reported endogenous ligand of GPR3[28], also stimulated a significant increase in glycolysis in ImKCs, although the magnitude of increase was much lower than that stimulated by 50 nM DPI (Fig. 2f), suggesting that activation of GPR3 by an endogenous ligand also stimulates glycolysis in macrophages.

β-arrestin2 has been reported to bind to GPR3 and is required for GPR3 signaling[29]. Consistently, a 10-min treatment with DPI, but not with S1P, stimulated the translocation of β-arrestin2 from the cytosol to the plasma membrane in both ImKCs and BMDMs (Fig. 2g and Supplementary Fig. 3a, b). To investigate the requirement for β-arrestin2 in DPI-stimulated glycolysis, we constructed *Arrb2*[–/–] ImKCs using CRISPR-Cas9 mediated gene editing (Supplementary Fig. 3c). Similar to *Gpr3*[–/–] BMDMs or *Gpr3*[–/–] ImKCs, the basal level of glycolysis and glycolytic capacity were significantly decreased in *Arrb2*[–/–] ImKCs

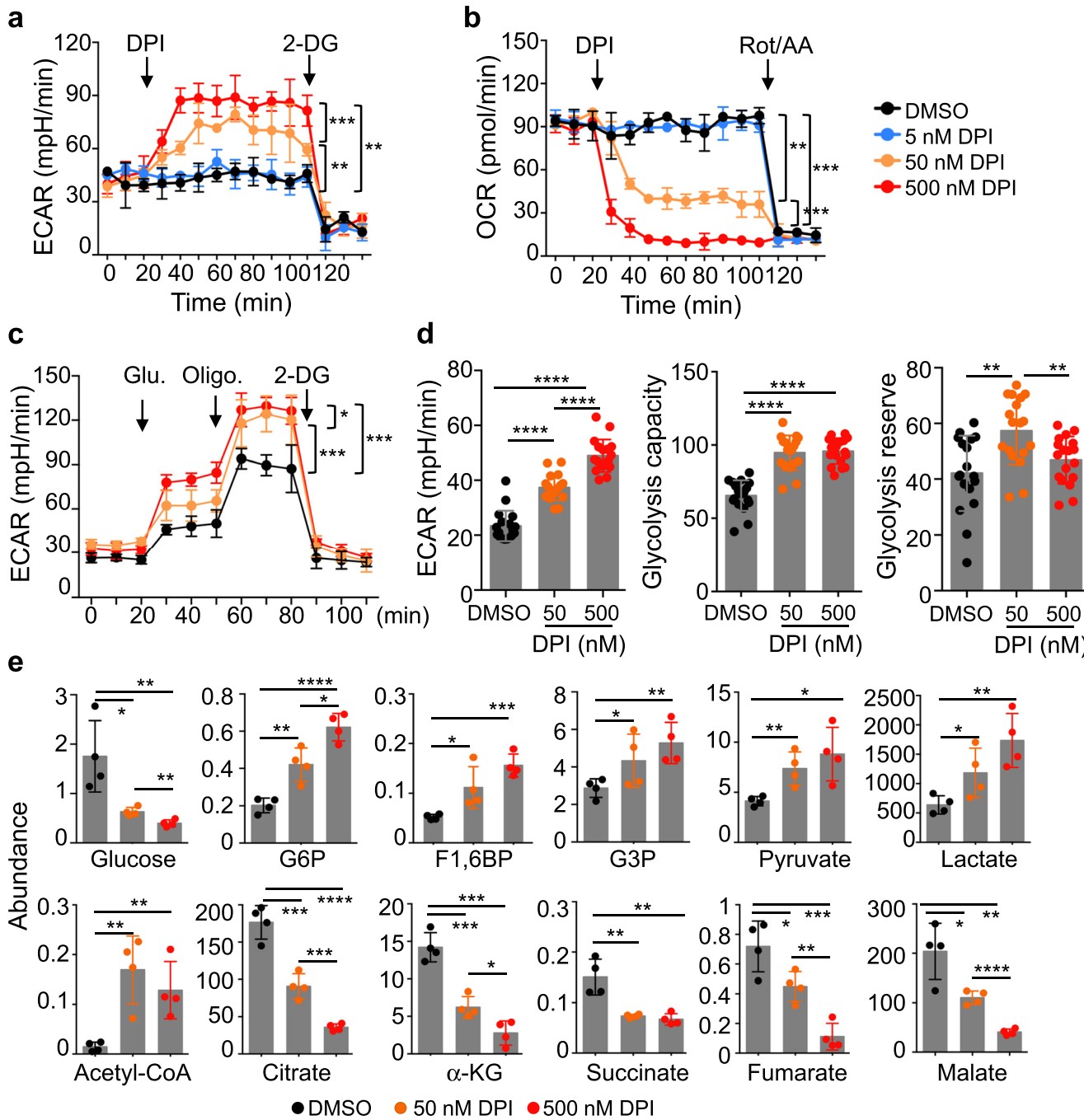

**Fig. 1 | DPI stimulates both rapid and sustained increases in glycolysis in macrophages. a, b.** The short-term effects of DPI on ECAR (**a**) and OCR (**b**) in ImKCs. ECAR and OCR were measured by Seahorse analyzer in ImKCs for 20 min, then for another 120 min following the addition of different concentrations of DPI (5, 50, or 500 nM), and then for another 40 min following the addition of 2-deoxylglucose (2-DG) (**a**) or rotenone plus antimycin A (Rot/AA) (**b**). Shown are representative data as the mean ± sd from three independent experiments. **c, d** The long-term effects of DPI on ECAR in ImKCs. ImKCs were seeded and incubated with or without DPI (50 and 500 nM) for 24 h. ECAR values were then measured under the basal conditions with sequential addition of 15 mM glucose, 2 μM oligomycin, and 50 mM 2-DG (**c**). Specific parameters for glycolysis, glycolytic capacity, and glycolytic reserve were calculated and data are presented as the mean ± sd from three independent experiments ($n = 18$ biological replicates) (**d**). **e** Select metabolite levels. ImKCs were treated with DPI for 6 h, and select metabolites in the glycolytic pathway and TCA cycle was quantified by LC−MS. Data are presented as the mean ± sd ($n = 4$ biological replicates). $P$ values were calculated by the two-sided student's $t$-test. *$P < 0.05$, **$P < 0.01$, ***$P < 0.001$, ****$P < 0.0001$. Source data are provided as a Source Data file.

as compared to the parental ImKCs (Fig. 2h, i and Supplementary Fig. 3d, e). Treatment with 50 nM DPI did not stimulate any significant increase in glycolysis, glycolytic capacity, and glucose uptake in *Arrb2*[−/−] ImKCs.

Together, these results show that DPI-stimulated glycolysis is dependent on GPR3 and β-arrestin2 and that activation of GPR3 by DPI leads to rapid trafficking of β-arrestin2 to the plasma membrane.

## DPI stimulates a rapid increase in glycolytic activity through the formation of GPR3-β-arrestin2 -GAPDH-PKM2 enzymatic super complex

How does DPI stimulate a rapid increase in glycolytic activity? We were intrigued by a previous report showing the interaction between β-arrestin2 and metabolic enzymes, including PK-3 (PKM1 or PKM2) and GAPDH, based on mass spectrometry analysis of

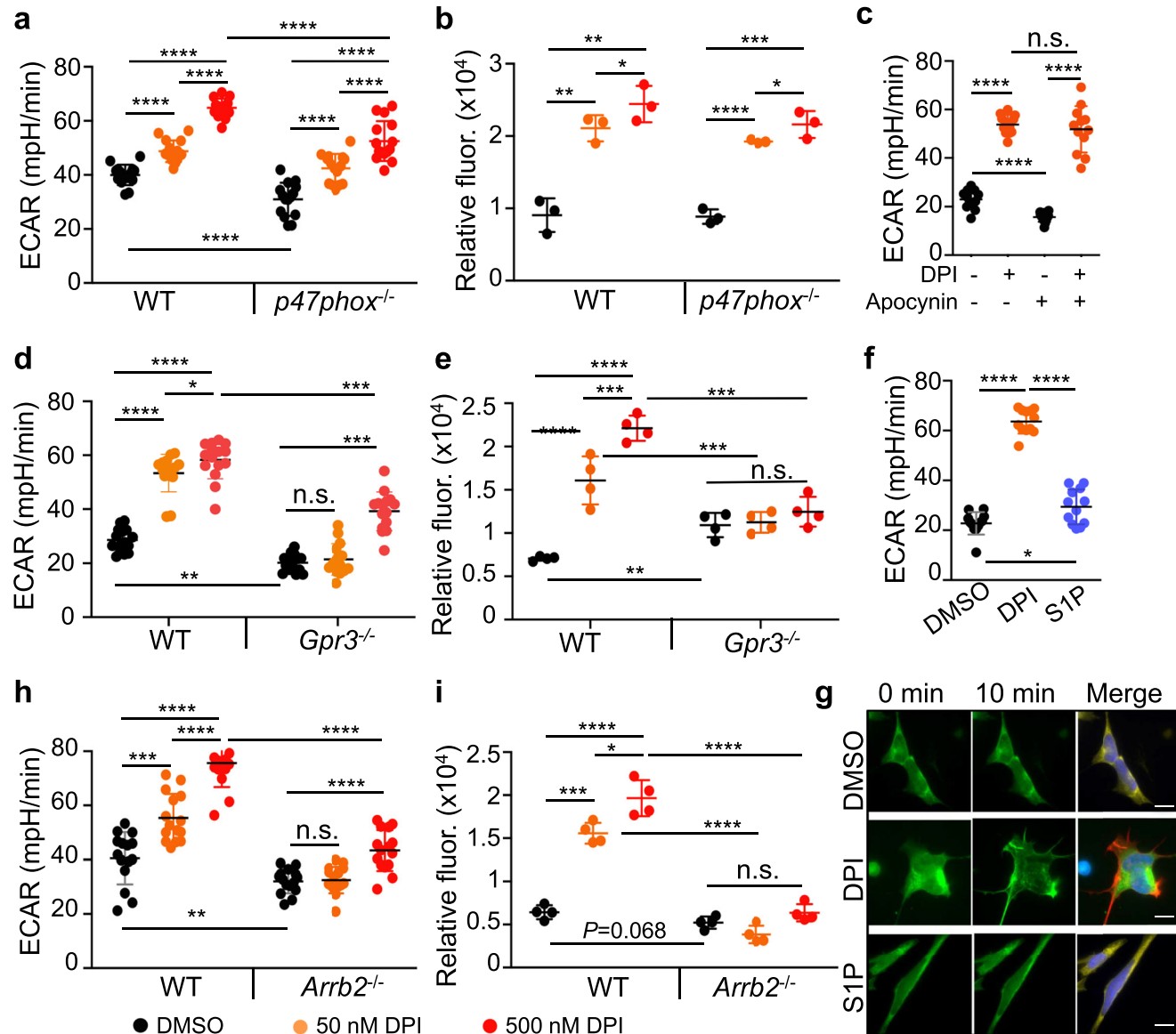

**Fig. 2 | DPI stimulates glycolysis through GPR3 and β-arrestin2. a** DPI-stimulated glycolysis is independent of NOX activity. Wildtype (WT) and *p47phox*⁻/⁻ BMDMs were treated with DMSO or DPI (50 and 500 nM) for 24 h and ECAR was measured by Seahorse analyzer. Data are presented as the mean ± sd from three independent experiments (*n* = 15 biological replicates). **b** The effect of DPI on glucose uptake in WT and *p47phox*⁻/⁻ BMDMs. BMDMs were treated with DMSO or DPI for 24 h in the presence of the fluorescent glucose analog 2-NBDG. The mean fluorescence intensity (MFI) of 2-NBDG in cells was measured by flow cytometry and normalized to DMSO controls. Data are presented as the mean ± sd (*n* = 3 independent experiments). **c** ECAR in WT BMDMs without or with DPI in the absence or the presence of the NOX inhibitor apocynin (100 µM) for 24 h. Data are presented as the mean ± sd from three independent experiments (*n* = 12 biological replicates). **d, e** DPI-stimulated glycolysis requires GPR3. ECAR (**d**) and glucose uptake (**e**) were measured in WT and *Gpr3*⁻/⁻ BMDMs without or with DPI treatment as described in

(**a**, **b**). *n* = 15 biological replicates in (**d**) from three independent experiments and *n* = 4 independent experiments in (**e**). **f** ECAR in ImKCs treated with DMSO, DPI (500 nM), or S1P (3 mM) for 24 h. Data are presented as the mean ± sd from three independent experiments (*n* = 12 biological replicates). **g** DPI induces β-arrestin2 translocation to cytoplasm membrane. ImKCs were transfected with the *Arrb2-GFP* fusion gene and stimulated with DMSO, DPI (50 nM), or S1P (3 mM). The GFP signal was captured with a TIRF microscope at indicated time points. Shown are representative data from three independent experiments. Scale bar: 5 µm. **h, i** DPI-stimulated glycolysis requires β-arrestin2. ECAR (**h**) and glucose uptake (**i**) were measured in WT and *Arrb2*⁻/⁻ ImKCs treated with DMSO or DPI. *n* = 15 biological replicates in (**h**) from three independent experiments and *n* = 4 independent experiments in (**i**). *P* values were calculated by the two-sided student's *t*-test. **P* < 0.05, ***P* < 0.01, ****P* < 0.001, *****P* < 0.0001. Source data are provided as a Source Data file.

anti-β-arrestin2 immunoprecipitates from HEK293 cells[30]. To investigate whether this mechanism might explain our metabolic findings, we treated ImKCs with or without DPI for 6 h and immunoprecipitated β-arrestin2, followed by Western blotting analysis. Consistent with the previous report[30], ERK1/2, enolase, GAPDH, and PKM2 (the PKM isoform expressed in the immune cells[31]) co-precipitated with β-arrestin2 (Fig. 3a). Notably, significantly higher levels of GAPDH and PKM2, but not PKM1 (the

PKM isoform expressed in many cell types), co-precipitated with β-arrestin2 following DPI treatment (Fig. 3a and Supplementary Fig. 4a), suggesting that DPI promotes interactions between β-arrestin2 and GAPDH and PKM2.

To determine the requirement for PKM2 in DPI-induced glycolysis, we treated BMDMs from wildtype mice and mice with deletion of exon 10 specific for the PKM2 isoform of PKM (*Pkm2*⁻/⁻)[31,32] with DPI and measured glycolytic activity. Similar to what was observed in

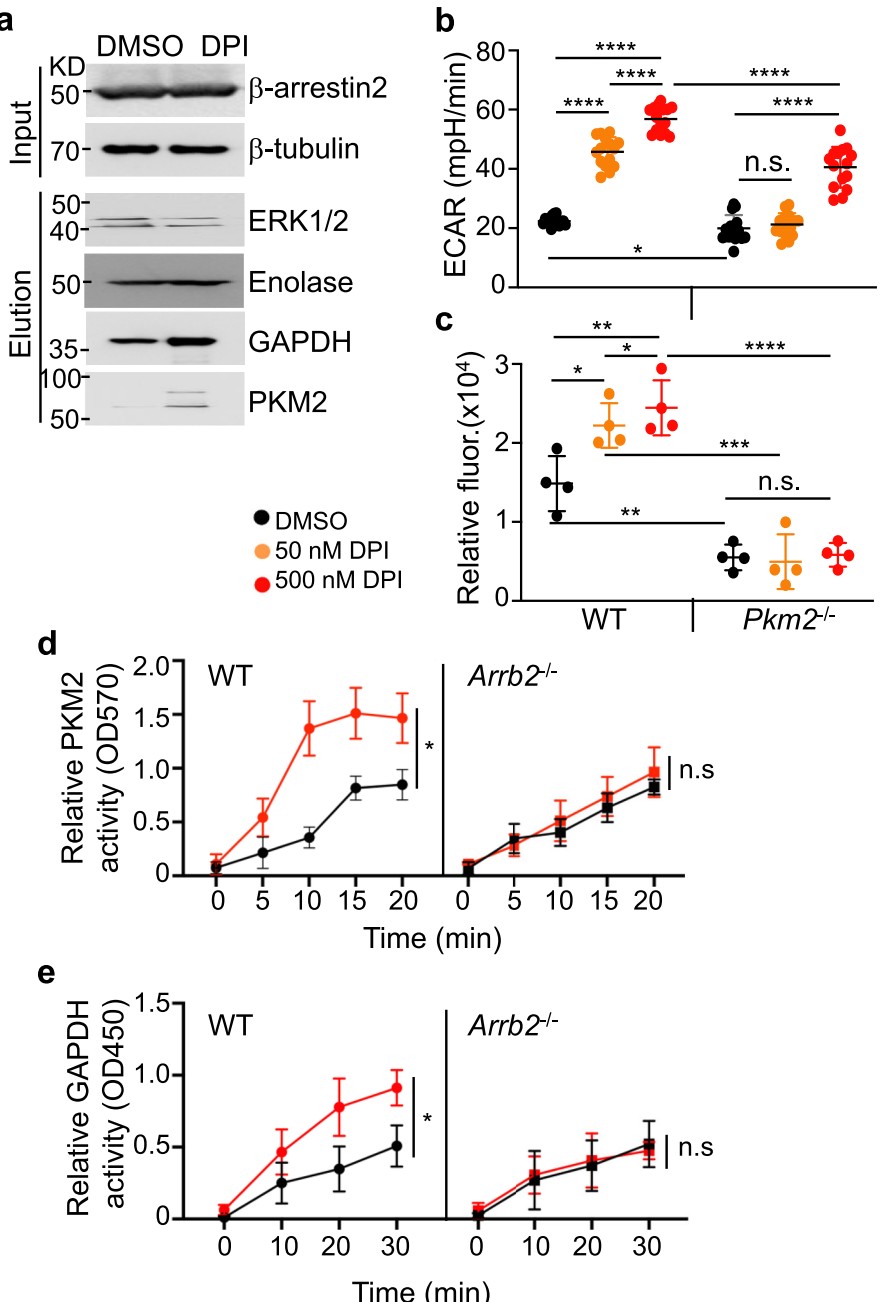

**Fig. 3 | DPI stimulates a rapid increase in glycolytic activity through the formation of GPR3-β-arrestin2-GAPDH-PKM2 enzymatic super complex. a** Co-IP of β-arrestin2 with ERK1/2, enolase, GAPDH, and PKM2. ImKCs were transfected with β-arrestin2 and then treated with or without 50 nM DPI for 6 h. Cell lysates were immunoprecipitated with anti-β-arrestin2 and the precipitates were analyzed by Western blotting for the indicated proteins. Shown are representative data from one of the three experiments. **b** DPI-stimulated glycolysis requires PKM2. BMDMs were prepared from wildtype and *Pkm2*^−/− mice, seeded, and incubated with or without DPI (50 and 500 nM) for 24 h, and ECAR was measured by a Seahorse analyzer. Data are presented as the mean ± sd from three independent experiments (n = 15 biological replicates). **c** WT and *Pkm2*^−/− BMDMs were seeded and incubated with or without DPI (50 and 500 nM) for 24 h in the presence of 2-NBDG to measure glucose uptake. Data are presented as the mean ± sd (n = 4 independent experiments). **d, e** DPI stimulates enzymatic activities of PKM2 and GAPDH. Wildtype and *Arrb2*^−/− ImKCs were treated with vehicle (black line) or 500 nM DPI (red line) for 6 h and the enzymatic activities of PKM2 (**d**) and GAPDH (**e**) were measured by colorimetric assay kits (Biovision). Data are presented as the mean ± sd from n = 3 independent experiments. P values were calculated by the two-sided student's t-test. * P < 0.05, ** P < 0.01, *** P < 0.001, **** P < 0.0001. Source data are provided as a Source Data file.

*Gpr3*^−/− BMDMs and *Arrb2*^−/− ImKCs, treatment of *Pkm2*^−/− BMDMs with 50 nM DPI did not stimulate any increase in glycolysis, glycolytic capacity and glucose uptake (Fig. 3b, c and Supplementary Fig. 4b, c). We also measured the enzymatic activity of PKM2 and GAPDH in the parental and *Arrb2*^−/− ImKCs in the absence or the presence of 500 nM DPI. DPI stimulated an immediate increase in PKM2 and GAPDH enzymatic activities in an β-arrestin2-dependent manner (Fig. 3d, e);

and the effect of DPI on PKM2 and GAPDH enzymatic activities was abolished when phosphorylation of ERK1/2 was inhibited by SCH772984 (Supplementary Fig. 4d, e). Consistently, DPI (50 nM) failed to stimulate any significant increase in glucose consumption and glycolytic intermediates of *Pkm2*^−/− BMDMs (Supplementary Fig. 4f). Thus, DPI stimulates the formation of GPR3-β-arrestin2-GAPDH-PKM2 complex and enhanced enzymatic activities of PKM2 and GAPDH,

providing a mechanistic explanation for the observed rapid increase in glycolytic activity following DPI treatment.

## DPI stimulates a sustained increase in glycolytic activity through nuclear translocation of PKM2 and transcriptional activation of c-Myc

We next examined the mechanism by which DPI stimulates the transcription of genes in the glycolysis pathway. PKM2 has been reported to form monomer, dimer, and tetramer. While the tetrameric form exhibits enzymatic activity, the dimeric form has been reported to translocate into the nucleus and function as a transcriptional cofactor to activate the expression of c-Myc[33–35], which can directly activate the transcription of almost all glycolytic genes through binding to the classical E-box sequence[36,37]. To test this mechanism in DPI-stimulated transcription of glycolytic genes, we prepared BMDMs from wildtype and *Pkm2*[-/-] mice, treated the cells with or without 50 and 500 nM DPI for 24 h, and then quantified the transcript levels of key glycolytic genes by RT-PCR. In a dose-dependent manner, DPI stimulated the transcription of *Pkm2*, *Ldha*, and *Hk2* in wildtype but not in *Pkm2*[-/-] BMDMs (Fig. 4a), suggesting PKM2 is required for mediating the DPI-stimulated transcription of glycolytic genes.

Next, we determined if DPI induces the formation of dimeric PKM2 and nuclear translocation. ImKCs were treated with 50 or 500 nM DPI for 1 h, lysed, and analyzed directly by native PAGE gel electrophoresis, followed by anti-PKM2 Western blotting. DPI treatment induced an increase of both the dimeric and tetrameric PKM2 but a decrease of the monomeric PKM2 (Fig. 4b, c). DPI treatment of ImKCs with longer time periods (6 and 12 h) induced the increase of all forms of PKM2 in a dose-dependent manner (Supplementary Fig. 5a). Induction of dimeric PKM2 by DPI was further confirmed by DSS crosslinking followed by Western blotting, and was abolished by inhibition of ERK1/2 with SCH772984 (Supplementary Fig. 5b, c), consistent with previous reports[34,38]. To further examine PKM2 nuclear translocation following DPI treatment, proteins from cytosolic and nuclear fractions were isolated from ImKCs that were treated with 50 nM DPI or DMSO for 6 h and analyzed by anti-PKM2 Western blotting. DPI treatment induced an increase in the levels of nuclear PKM2 (Fig. 4d). Moreover, both ImKCs and human primary KCs were treated with DPI or vehicle for 24 h and then stained directly with anti-PKM2. In the absence of DPI treatment, anti-PKM2 fluorescent signals were localized exclusively in the cytosol, whereas with DPI treatment, a significant amount of anti-PKM2 fluorescent signals was detected in the nucleus (Fig. 4e), indicating that DPI treatment results in the translocation of PKM2 from the cytosol into the nucleus.

We determined if c-Myc expression is induced by DPI treatment in a PKM2-dependent manner. As shown in Fig. 4a, DPI stimulated the transcription of *c-Myc* in wildtype but not *Pkm2*[-/-] BMDMs in a dose-dependent manner. To determine whether DPI activates c-Myc transcriptional activity, we performed c-Myc luciferase reporter assays in the wildtype and *Pkm2*[-/-] BMDMs. Luciferase activity was induced by DPI only in wildtype but not in *Pkm2*[-/-] BMDMs (Fig. 4f), suggesting that DPI stimulates c-Myc transcription and activates subsequent c-Myc transcriptional activity in a PKM2-dependent manner.

Taken together, these results indicate that DPI stimulates a sustained increase in glycolytic activity through nuclear translocation of PKM2, transcriptional activation of c-Myc, and transcription of glycolytic genes.

## DPI inhibits HFD-induced obesity and liver pathogenesis

To examine the effect of DPI-induced glycolysis in vivo, we measured the fasting glucose response in DPI-treated mice. C57BL/6 (B6) mice were injected intraperitoneally (i.p.) with 2 mg/kg DPI, and 6 h later, the mice were injected i.p. with 1 g/kg glucose. Blood glucose levels were measured before DPI injection, 6 hours after DPI injection, and at different time points after glucose injection. As shown in

Supplementary Fig. 6a, mice had the same levels of blood glucose before DPI injection. Six hours after DPI injection, DPI-treated mice had significantly lower levels of blood glucose and maintained significantly lower levels of glucose 15 and 30 min after glucose injection, suggesting that DPI stimulates an increased rate of glucose consumption in vivo.

To investigate the effect of DPI at the organismal level, we examined whether DPI could inhibit HFD-induced obesity and liver pathogenesis. B6 mice at 5 weeks of age were fed with HFD or normal chow (NC) for a total of 8 weeks. Three weeks after the start of HFD, when the mice had already exhibited significant weight gain, the mice were given either vehicle (PEG3000) or DPI (2 mg/kg) in vehicle i.p. every 5 days for a total of 6 doses. In the HFD-fed mice, DPI treatment significantly reduced the weight gain as compared to the vehicle-treated group (Fig. 5a) without affecting the weekly food intake (Fig. 5b). Consistently, DPI-treated mice had significantly lower amount of eWAT after 8 weeks of HFD (Fig. 5c). Notably, DPI-treated HFD mice gained weight at a rate comparable to mice fed with a NC (Fig. 5a), suggesting that DPI appears to inhibit the weight gain due to extra fat uptake from HFD without impairing normal growth. Glucose tolerance tests showed that the DPI-treated HFD mice displayed a significant increase in glucose tolerance compared to the vehicle-treated HFD mice (Fig. 5d), consistent with significantly reduced levels of blood insulin (Supplementary Fig. 6b). DPI treatment dramatically reduced the levels of lipid deposition and triacylglycerol (TAG) in the liver, the levels of serum ALT and AST, and the size of adipocytes in eWAT as compared to vehicle-treated HFD mice (Fig. 4e, f and Supplementary Fig. 6c, d).

We also examined the effect of DPI on hepatic steatosis by feeding B6 mice with HFD for 16 weeks. Beginning nine weeks after HFD initiation, when the mice had become obese, DPI (2 mg/kg) was given once every 5 days for a total of 10 doses. As expected, DPI also significantly reduced the weight gain without affecting the weekly food intake (Supplementary Fig. 6e, f). The weight of eWAT was significantly lower in the DPI-treated group than in the vehicle-treated group (Supplementary Fig. 6g). Similarly, the DPI-treated HFD mice displayed an increased glucose tolerance and had reduced lipid droplet, steatosis, and collagen deposition in the liver (Supplementary Fig. 6h, i). Together, these results show that DPI inhibits HFD-induced obesity, lipid deposition, and hepatic steatosis in mice.

## DPI inhibits HFD-induced pathologies through PKM2 expression in Kupffer cells

In humans and mice, pyruvate kinases are encoded by two genes (*PKLR* and *PKM*), producing four isoforms: *PKR*, *PKL*, *PKM1* and *PKM2*. *PKR* is expressed exclusively in erythrocytes[31] and *PKL* is expressed predominantly in hepatocytes, while *PKM1* is expressed in many non-lymphoid tissues, and *PKM2* is predominantly expressed in lymphoid tissues. To investigate the cell types in the liver that mediate the effect of DPI treatment, we analyzed the expression of *PKL*, *PKM1*, and *PKM2* in different cell types in the livers using published single-cell RNAseq data[13,39]. In both humans and mice, *PKM2* was highly expressed in KCs and intermediately expressed in other immune cells but minimal in APOC3+ hepatocytes, whereas *PKM1* was detected at much lower levels in human primary macrophages and mouse KCs (Supplementary Fig. 7a, b). DPI treatment of ImKCs upregulated the expression of *PKM2* but not *PKM1* (Supplementary Fig. 7c). As expected, *PKL* was exclusively expressed in APOC3+ hepatocytes. Consistently, DPI at 50 nM did not increase ECAR in mouse primary hepatocytes and adipocyte-like cells differentiated from 3T3-L1 cells, while 500 nM DPI had some effect (Supplementary Fig. 8a, b), similar to the results obtained with *Gpr3*[-/-], *Arrb2*[-/-] and *Pkm2*[-/-] macrophages.

To directly test whether PKM2 expression in KCs mediates the effect of DPI, we constructed KC-specific PKM2 knockout (*Pkm2*[-/-])

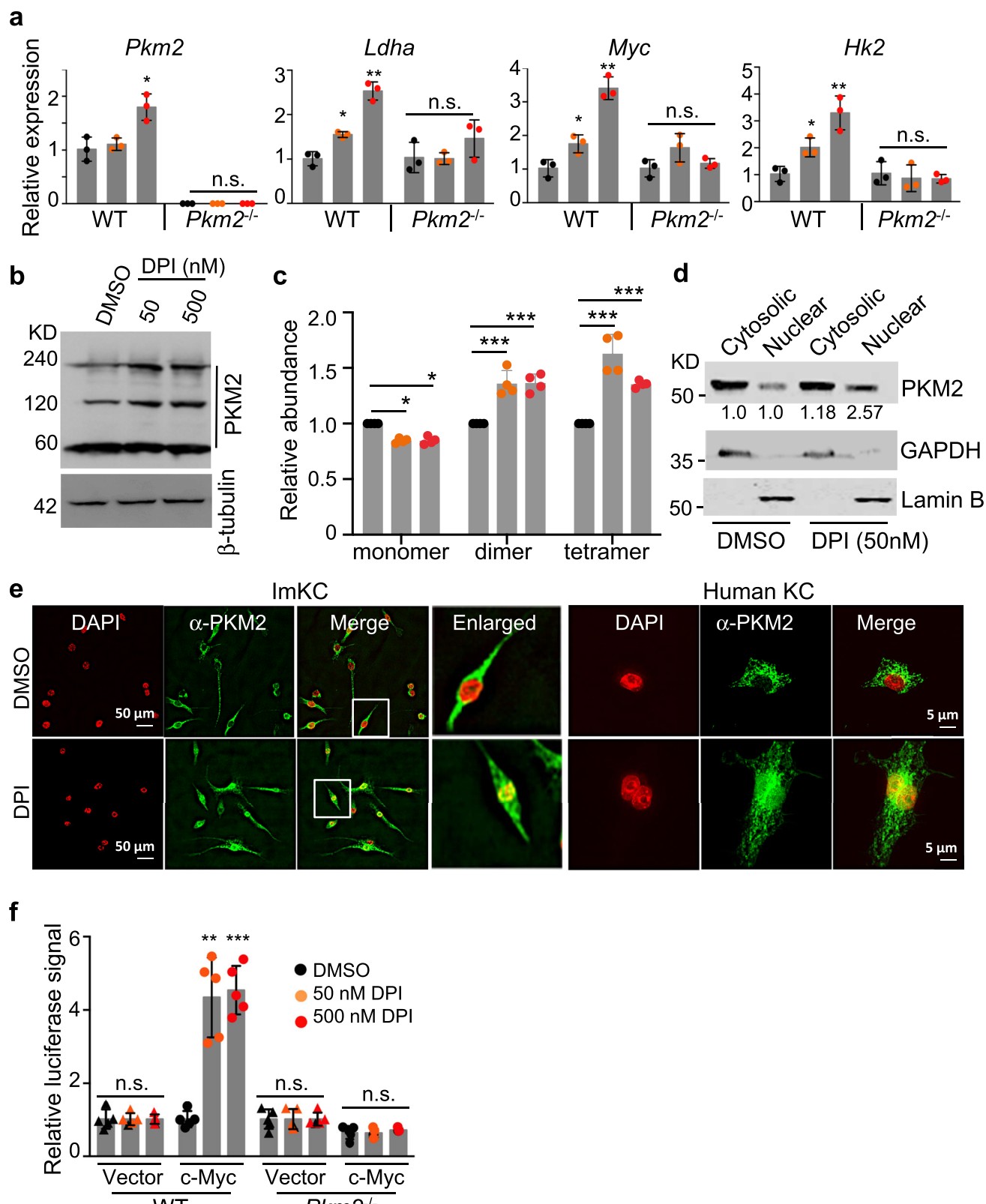

mice by crossing Clec4f-Cre mice with exon 10 floxed *Pkm2f/f* mice. KC-specific *Pkm2−/−* mice, as well as control *Pkm2f/f* mice, were fed with HFD or NC for 8 weeks starting at 5 weeks of age. Three weeks after HFD, the mice were given vehicle or DPI (2 mg/kg) i.p. every 5 days for a total of 6 doses. Both *Pkm2−/−* and *Pkm2f/f* mice gained significantly more weight on HFD than on NC, and the gain was inhibited by DPI treatment in *Pkm2f/f* mice but not *Pkm2−/−* mice (Fig. 5g and Supplementary Fig. 9a),

although the mice had similar food intake (Supplementary Fig. 9b). Consistently, DPI treatment of HFD-fed *Pkm2−/−* mice failed to increase glucose tolerance and reduce the levels of eWAT, lipid deposition and TAG in the liver, serum ALT and AST, and blood insulin as compared to HFD-fed *Pkm2f/f* mice (Fig. 5h and Supplementary Fig. 9c−g). These results show that DPI inhibition of HFD-induced obesity and liver pathogenesis requires PKM2 expression in KCs.

**Fig. 4 | DPI stimulates a sustained increase in glycolytic activity through nuclear translocation of PKM2 and transcriptional activation of c-Myc. a** DPI-induced transcription of glycolytic genes requires PKM2. WT and *Pkm2*[−/−] BMDMs were treated with DPI (50 and 500 nM) or vehicle for 24 h. The transcript levels of *Pkm2*, *Ldha*, *Hk2*, and *c-Myc* were measured by real-time qPCR. Data were collected from two independent experiments with three biological replicates per group. Transcriptional level was normalized to *β-actin* first and then to DMSO control. Data are presented as the mean ± sd. **b**, **c** Induction of dimeric PKM2 by DPI. ImKCs were treated with vehicle or DPI (50 and 500 nM) for 1 h. Cell lysates were run on native PAGE gel and analyzed by Western blotting. Shown are representative data (**b**) and summarized data (**c**) quantified by ImageJ from three independent experiments. **d** DPI induces nuclear translocation of PKM2 by Western blotting. ImKCs were treated with vehicle or DPI (50 nM) for 6 h. Proteins from cytosolic and nuclear fractions were isolated and analyzed by anti-PKM2 Western blotting. Shown are representative data, and the numbers are average expression levels of cytosolic and nuclear PKM2 from three independent experiments. **e** DPI induces nuclear translocation of PKM2 by confocal microscopy. ImKCs and human primary KCs (Acce-Gen) were treated with vehicle or DPI (50 nM) for 24 h, stained with anti-PKM2 (green) and DAPI (red), followed by confocal imaging. Shown are representative images from two independent experiments. The boxed areas are enlarged and scale bars are indicated. **f** DPI stimulates transactivation of c-Myc. c-Myc luciferase reporter plasmid was transfected into WT and *Pkm2*[−/−] BMDMs. Transfected cells were treated with vehicle or DPI (50 and 500 nM) for 6 h, and luciferase activities were measured. Data are presented as the mean ± sd from two independent experiments (*n* = 5 biological replicates). *P* values were calculated by the two-sided student's *t*-test. \*P < 0.05, \*\*P < 0.01, \*\*\*P < 0.001, \*\*\*\*P < 0.0001. Source data are provided as a Source Data file.

We also directly examined the effect of DPI on glycolysis of freshly isolated KCs from the livers of B6 mice, *Pkm2*[f/f] mice, and KC-specific *Pkm2*[−/−] mice. DPI at 50 nM and 500 nM stimulated a significant increase in ECAR, glycolytic capacity, and glycolytic reserve in B6 KCs in a dose-dependent manner (Supplementary Fig. 10a, b) to a similar magnitude as observed in both BMDMs and ImKCs (Supplementary Fig. 2). Similarly, DPI at 50 nM and 500 nM stimulated a significant increase in ECAR, glycolytic capacity and glycolytic reserve in *Pkm2*[f/f] KCs in a dose-dependent manner (Supplementary Fig 10c, d). However, DPI at 50 nM failed to stimulate any significant increase in ECAR, glycolytic capacity, and glycolytic reserve in *Pkm2*[−/−] KCs, similar to *Gpr3*[−/−] or *Arrb2*[−/−] ImKCs and *Gpr3*[−/−] BMDMs (Supplementary Fig 10c, d, Fig. 2 and Supplementary Figs. 2 and 3). Furthermore, we purified KCs from vehicle- or DPI-treated HFD-fed *Pkm2*[f/f] and KC-specific *Pkm2*[−/−] mice and directly measured their basal level of ECAR. ECAR was significantly higher in KCs from *Pkm2*[f/f] mice that were treated with DPI than in vehicle (Fig. 5i). In contrast, ECAR was significantly lower in KCs from *Pkm2*[−/−] mice, and DPI treatment did not increase ECAR. These results show that DPI likely stimulates glycolysis in KCs in mice through the same mechanisms as in BMDMs and ImKCs.

## DPI stimulates the expression of glycolytic genes but suppresses the expression of inflammatory genes in KCs from HFD-fed mice

To further investigate the effects of DPI on KCs, we purified KCs from vehicle- or DPI-treated HFD-fed mice and age-matched mice fed with an NC and performed RNA-seq. GSEA and functional enrichment analysis showed the upregulation of genes associated with immune and inflammatory responses in KCs from mice fed with HFD compared to those fed an NC (Fig. 6a–c and Supplementary Data 1 and 2), consistent with previous reports[10,13]. The expression of genes involved in inflammation was significantly suppressed in KCs from HFD mice following DPI treatment. In contrast, DPI treatment upregulated the expression of many other genes that were downregulated in KCs from HFD mice (Fig. 6a). For example, the expression of genes involved in glycolysis, oxidative phosphorylation, and fatty acid metabolism was downregulated in KCs from HFD-fed mice, whereas their expression was upregulated in KCs from HFD-fed mice following DPI treatment (Fig. 6a–c). Macrophage polarization index (MPI)[40] analysis showed that KCs were polarized to an M1-like state in HFD-fed mice but to an M2-like state in mice fed with a normal chow, while KCs were reprogrammed to an intermediated phenotype in DPI-treated HFD mice (Fig. 6d).

We also directly assayed the effect of DPI on the production of inflammatory cytokines and reactive oxygen species (ROS) in purified KCs from vehicle- or DPI-treated HFD-fed mice and age-matched mice fed with an NC. Purified KCs were not stimulated or stimulated with LPS for 6 h, cytokines in the culture supernatants were assayed by ELISA, and intracellular ROS were detected by CM-H2DCFDA staining. LPS stimulated the production of pro-inflammatory cytokines TNF-α, IL-1β, and IL-6 in KCs, but the levels were significantly lower in KCs from DPI-treated mice than those of vehicle-treated mice, reaching similar levels as in KCs from mice fed with NC (Fig. 6e). Interestingly, the basal levels of these cytokines were also lower in KCs from DPI-treated mice than vehicle-treated mice. Similarly, LPS-induced ROS production was significantly reduced in KCs purified from DPI-treated mice as compared to vehicle-treated mice (Fig. 6f).

We also purified KCs from normal B6 mice and treated them with DMSO, 50 nM, or 500 nM DPI in vitro for 24 h, followed by LPS stimulation in fresh medium without DPI for 6 h. DPI pre-treatment significantly inhibited LPS-stimulated production of TNF-α, IL-1β, IL-6, and ROS (Supplementary Fig 11a, b). Moreover, we broadly assayed the levels of cytokines and chemokines in the plasma of normal B6 mice following DPI treatment (2 mg/kg, i.p.) for 48 h using a mouse cytokine antibody array specific for 62 targets (Abcam, #Ab133995). Compared to the plasma collected before DPI treatment, only two anti-inflammatory cytokines, CCL9 and CXCL4, were visibly elevated following DPI treatment; none of the proinflammatory cytokines were elevated (Supplementary Fig. 11c). These results indicate that DPI stimulates glycolysis but suppresses inflammatory responses of KCs in HFD-fed mice, consistent with reduced pathogenesis in the liver.

## DPI stimulates the expression of glycolytic genes but suppresses the expression of inflammatory genes in KCs from patients with NAFLD

Single-cell RNAseq analysis of liver cells from NASH and cirrhosis patients has identified the presence of TREM2[+] disease-associated macrophages (DAMs) in the liver that display lower expression of metabolic genes[12,13]. To determine whether the DAMs are also present in patients with mild NAFLD, we performed scRNAseq of freshly isolated immune cells from liver biopsies of three healthy donors and three mild NFALD patients (Supplementary Fig. 12a, b). Fourteen cell clusters were identified, including naïve CD8[+] T cells, resident memory CD8[+] (T_RM) cells, CD4[+] T cells, B and plasma cells, CD56[low] and CD56[hi] NK cells, macrophages or KCs, neutrophils and proliferating cells (Supplementary Fig. 12c, d). Cellular composition analysis identified only macrophages or KCs were increased significantly in NAFLD patients compared to the healthy donors (Supplementary Fig. 12e, f). Functional analysis of differentially expressed genes between NAFLD and healthy livers showed that T_RM and NK cells were more activated (Supplementary Fig. 12g).

To further decode the changes in macrophages, we re-clustered 5497 CD14[+]CD68[+] liver macrophages (LM1, LM2 and LM3) into 7 sub-populations (Fig. 7a and Supplementary Fig. 13a). Although no difference was observed in cell numbers of each subcluster between NAFLD and healthy donors (Supplementary Fig. 13b), GO enrichment analysis of DEGs identified between NAFLD and healthy donors showed an overall upregulation of pathways of antigen presentation and inflammatory response (Supplementary Fig. 13c).

These subpopulations were annotated based on previous publications[11,12,39]. Cluster 1 (C1, 22.6%) and C2 (21.7%) were monocyte-

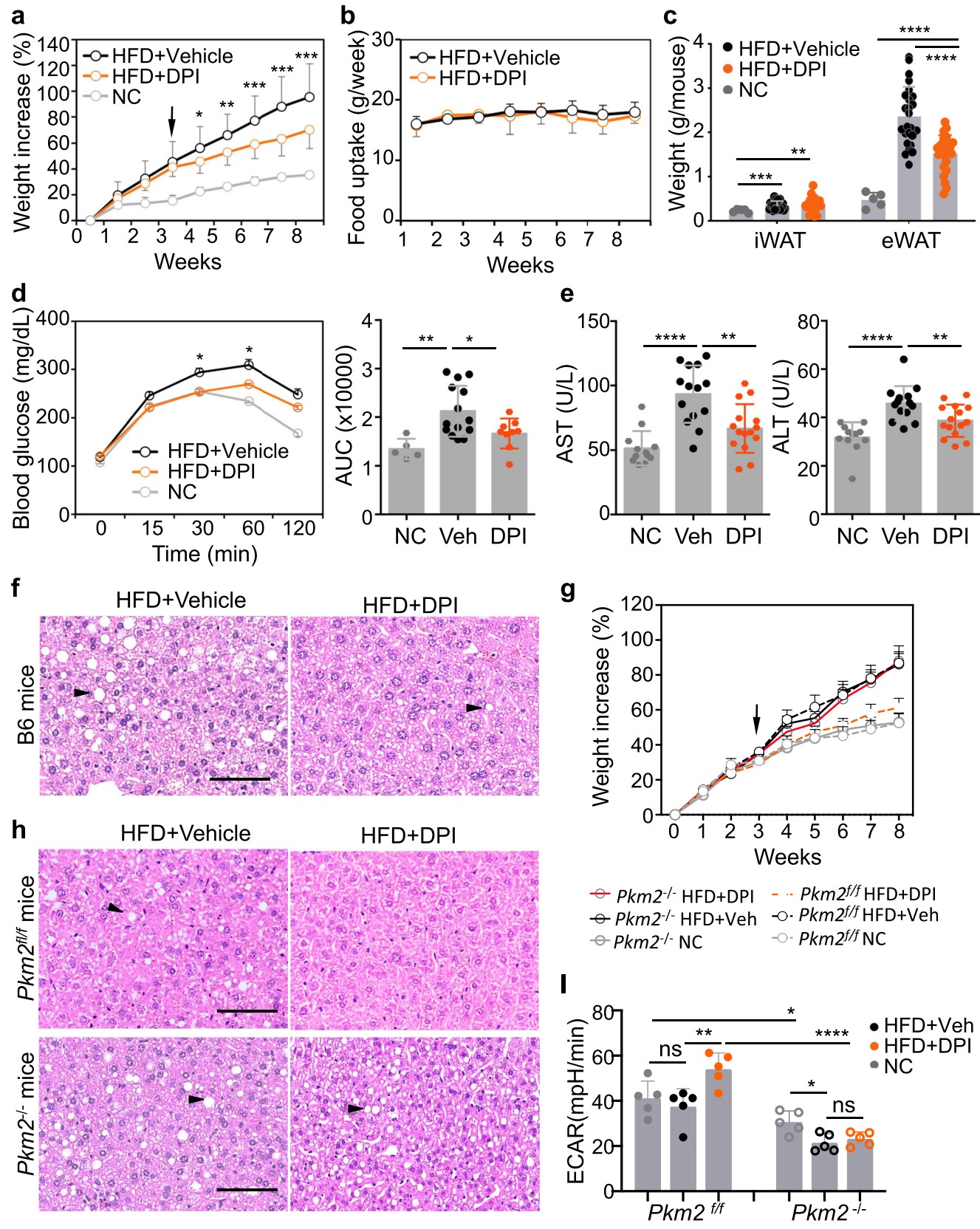

derived liver macrophages (moMac) as they expressed a high level of *MNDA* but a low level of *MACRO* and *CD163*[12]. C1 differed from C2 by expressing higher levels of inflammatory genes (Supplementary Fig. 13d), whereas C2 expressed higher levels of glycolytic genes, including *PGAM1, PKM, GAPDH,* and *ENO1* (Fig. 7b), indicating two

moMac populations existed in human liver with distinguished inflammatory and metabolic status. C0 (24.1%), C3 (13.2%), and C4 (9.4%) all expressed KC markers *CD163* and *MACRO*. The expression profile of C4 was likely dendritic cells, as some cells expressed *CD1C*. The expression profile of C3 resembled that of DAMs based on expression of *APOE*,

**Fig. 5 | DPI inhibits HFD-induced obesity and liver pathogenesis through PKM2 expression in Kupffer cells. a–f** DPI prevents weight gain in mice fed with HFD. Male B6 mice at 5 weeks of age were fed with HFD or normal chow (NC) for a total of 8 weeks (week 8). Three weeks after HFD (arrow), the mice were given either vehicle (Veh) or DPI (2 mg/kg) every 5 days for a total of 6 doses. The body weight (**a**) and food consumption (**b**) were monitored weekly. Data are presented as the mean ± sd from three independent experiments (*n* = 5–10 mice per group per experiment). **c** The weights of eWAT and iWAT at week 8. **d** Fasting glucose tolerance assay. At week 7 plus 3 days, mice were starved overnight (12–16 h) with water ad libitum. Glucose (1 g/kg) was injected i.p. and blood glucose levels were monitored at the indicated time. AUC (right panel) was calculated for statistics. **e** Serum AST and ALT activities in mice at week 8. **f** Comparison of H&E staining of liver sections at week 8. Shown are representative H&E staining from one mouse per group from (**a**). **g–i** Effect of DPI on KC-specific *Pkm2*$^{-/-}$ mice fed with HFD. Experimental

procedures with male KC-specific *Pkm2*$^{-/-}$ mice and control *Pkm2*$^{f/f}$ mice were described in (**a**). Body weights were monitored weekly (**g**). Data are presented as the mean ± sd from two independent experiments with *n* = 5 mice per group per experiment. *P* values were calculated by the two-sided student's *t*-test and shown in Supplementary Fig. 9a. Mice fed with NC are used as controls. Statistical *P*-values between groups were shown in Supplementary Fig. 9a. Comparison of H&E staining of liver sections at week 8 (**h**). Shown are representative H&E staining from one mouse per group. Black arrows in **f** and **h** point to lipid droplets. Scale bar: 100 μm. F4/80$^+$ Liver macrophages were purified from mice in **g** at week 8 using anti-F4/80 microbeads. ECAR was measured directly by Seahorse analyzer (**i**). Data are presented as the mean ± sd (*n* = 5). Each dot represents one mouse in (**c–e**) and (**i**). *P* values were calculated by the two-sided student's *t*-test. \**P* < 0.05, \*\**P* < 0.01, \*\*\**P* < 0.001, \*\*\*\**P* < 0.0001, ns not significant. Source data are provided as a Source Data file.

*TREM2, CD9, GPNMB,* and *CLEC10A,* as well as complement genes (C1QA, etc.). GO enrichment analysis of DEGs between C3 and C0 or C3 and C1/2 showed that C3 upregulated pathways of antigen processing and presentation, monocyte chemotaxis, chemokine-mediated signaling, and down-regulated pathways of the immune response, glycolysis, phagocytosis (Fig. 7d, e), as observed in advanced NASH and cirrhosis[12,13]. Based on the trajectory inference (Fig. 7c) and enriched GO ontology pathways (Fig. 7d, e and Supplementary Fig. 13d), C0 likely represents KCs, and C1 represents the intermediate or differentiating LM between resident KCs (C0) and moMac (C2), suggesting KCs are likely repopulated from moMac (C1) in the liver. DAMs (C3) might be developed from KCs (C0) as evidenced by co-expression of multiple genes, including *MARCO, CD163, LIPA, CCL3, CCL4,* and *CXCL3* (Fig. 7b and Supplementary Fig. 13). C5 (5.9%) expressed high levels of myeloid checkpoint receptors LILRB1 and LILRB2. C6 (3.1%) likely corresponds to KCs phagocytosing red blood cells based on co-expression of hemoglobin mRNAs (HBD and HBA2)[41] (Fig. 7b and Supplementary Fig. 13d).

To directly examine the effect of DPI on human KCs from NFALD patients, we purified KCs from two additional NAFLD patients and performed the transcriptional analysis by RNA-seq following DPI treatment ex vivo for 24 h. As seen in human MDMs and mouse ImKCs, the expression of glycolytic genes was upregulated by DPI treatment, whereas the expression of DAM markers, including APOE, CLEC10A, TREM2, and C1QA, was downregulated (Fig. 7f). Functional enrichment analysis showed that DPI-treated KCs not only upregulated the expression of glycolytic genes but also suppressed the expression of genes associated with chemokine-mediated signaling, chemotaxis and inflammatory response (Fig. 7g and Supplementary Data 3). DPI (50 nM) did not stimulate the secretion of proinflammatory cytokines and chemokines in hMDMs differentiated from human monocytes from healthy donors at the protein level (Supplementary Fig. 11d). In contrast, DPI (50 nM and 500 nM) suppressed the production of intracellular ROS in hMDMs (Supplementary Fig. 11e), and enhanced hMDM phagocytosis of *Escherichia coli* bioparticles (Supplementary Fig. 11f). These results indicate that DPI also upregulates expression of glycolytic genes and suppresses expression of inflammatory genes in KCs from patients with NAFLD.

## Discussion

In this study, we investigated the effects of GPR3 activation by DPI on metabolic reprogramming at the cellular and organismal levels and elucidated the underlying molecular mechanisms. DPI has been reported as an agonist of GPR3 and an inhibitor of NOX enzyme[25,26]. Consistent with previous observation that NOX-deficiency leads to a lower cellular glycolysis[42], we found that *p47phox*$^{-/-}$ BMDMs and inhibition of NOX activity by apocynin in macrophages lead to a significantly reduced basal level of glycolytic activity. However, DPI (50 nM) stimulated a similar level of increase in ECAR in *p47phox*$^{-/-}$ BMDMs as in wildtype BMDMs, or with or without inhibitor apocynin,

suggesting that DPI stimulates glycolysis independent of its NOX inhibiting activity. In contrast, although GPR3 knockout also reduces the basal level of glycolytic activities, DPI (50 nM) failed to stimulate any significant increase in ECAR, suggesting that DPI stimulates glycolysis through the activation of GPR3. Similarly, β-arrestin2 and PKM2 are required for mediating the effect of DPI on glycolysis as knockout of these genes in BMDMs or ImKCs or primary KCs abolishes DPI-induced increases in ECAR. At 500 nM, DPI stimulated an increase in glycolytic activities in macrophages with GPR3 β-arrestin2 or PKM2 knockout. This is likely due to the activation of other molecules by DPI at such a high concentration. These genetic analyses identify a pathway involving GPR3, β-arrestin2, and PKM2 in mediating the effect of DPI on glycolysis as well as NOX, GPR3, β-arrestin2, and PKM2 in maintaining a threshold level of basal cellular glycolysis. As SIP, a putative endogenous ligand of GPR3, also induces a significant increase in glycolysis in macrophages, the identified pathway likely functions in the metabolic reprogramming of macrophages under physiological conditions.

Consistent with the critical role of β-arrestin2 in GPCR signaling by functioning as a scaffold protein[29], we show that activation of GPR3 by DPI leads to a rapid recruitment of β-arrestin2 but not β-arrestin1 to the plasma membrane, presumably to GPR3[29]. Biochemically, we show that activation of GPR3 by DPI results in the formation of a glycolytic enzyme super complex, including β-arrestin2, enolase, GAPDH, and PKM2, in an ERK1/2-dependent manner, consistent with previous report in HEK293 cells[43]. Associated with the formation of the glycolytic enzyme super complex, the enzymatic activities of GAPDH and PKM2 were greatly elevated, providing a biochemical basis for the rapid increase of glycolytic activities following DPI treatment.

We found that activation of GPR3 by DPI also promotes the formation of dimeric PKM2 in an ERK1/2-dependent manner. Dimeric PKM2 has been reported to translocate from the cytosol to the nucleus, where it transactivates c-Myc[34,38]. c-Myc is known to directly activate the transcription of almost all glycolytic genes through binding the classical E-box sequence[36,37]. We found that DPI stimulates PKM2 translocation into the nucleus in both ImKCs and primary human KCs. DPI also stimulates both transcription of c-Myc in a PKM2-dependent manner and c-Myc transactivation activities using reporter gene assays. These findings provide a molecular mechanism by which DPI stimulates a sustained increase in glycolysis in macrophages through the transcription of glycolytic genes.

Consistent with the increased glucose consumption through enhanced glycolysis at the cellular level, DPI has profound effects on glucose metabolism and on HFD-induced weight gain, lipid deposition, and fibrosis in the liver at the organismal level. DPI confers a better glucose tolerance in mice under normal conditions. DPI significantly inhibits HFD-induced weight gains without affecting food intake, consistent with the development of late-onset of obesity in *Gpr3*$^{-/-}$ mice and *Pkm2*$^{-/-}$ mice[18,19]. Impressively, DPI treatment of obese mice on HFD every 5 days is sufficient to almost eliminate lipid droplet

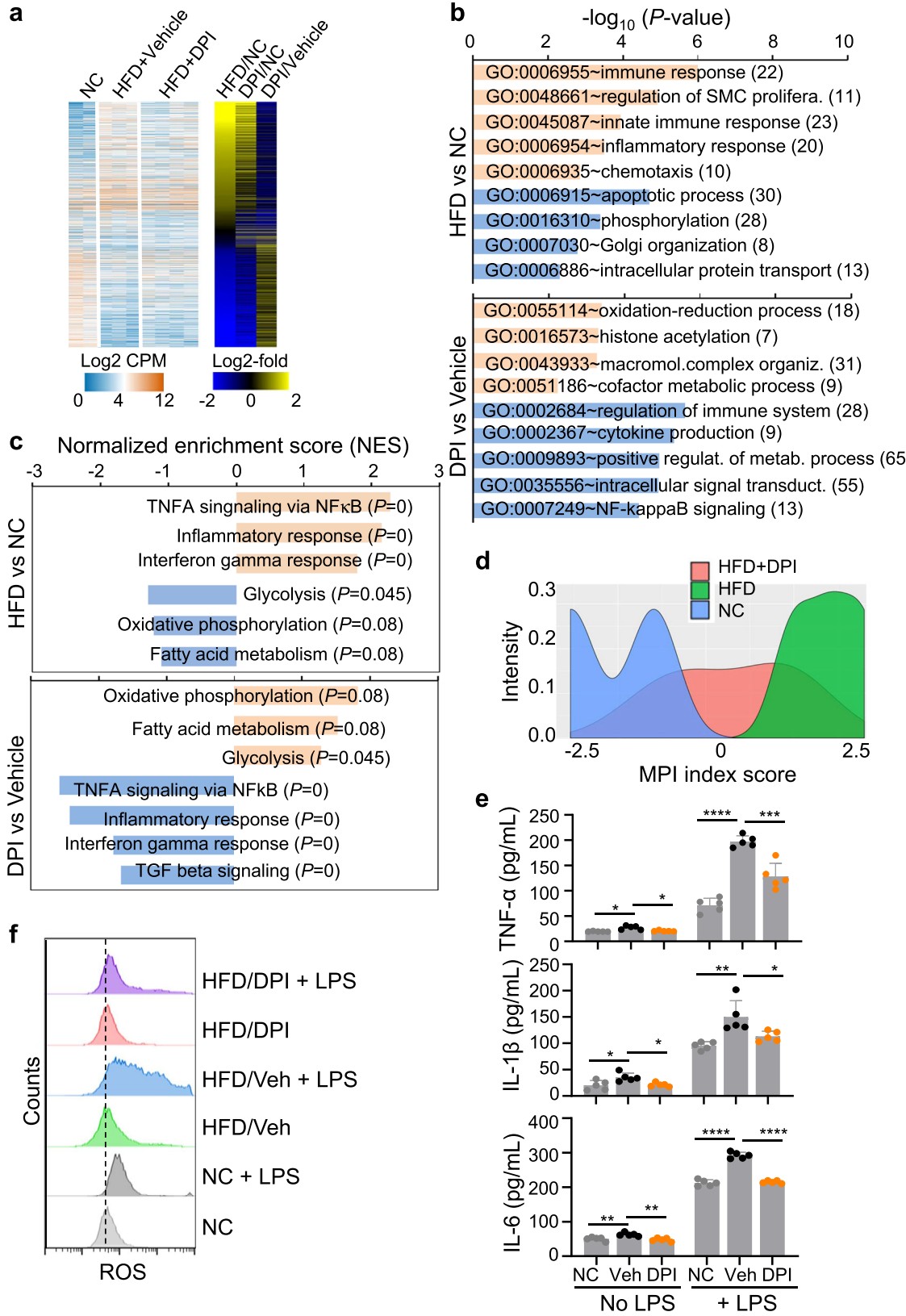

accumulation and fibrosis in the liver, suggesting that DPI's effect on liver pathologies is not completely dependent on body weight reduction. Mechanistically, DPI greatly stimulates the expression of genes in the glycolysis pathway but suppresses the expression of inflammatory genes in KCs from HFD-fed mice (Fig. 6), suggesting the effect on liver pathologies is likely a result of both increased glycolysis (and therefore reduced lipid accumulation) and reduced inflammation (fibrosis). Similarly, DPI treatment of KCs from NAFLD patients upregulated the glycolysis pathway and response to hypoxia but downregulated pro-inflammatory pathways at the transcription level (Fig. 7f, g). At the protein level, DPI inhibited LPS-stimulated production of proinflammatory cytokines and ROS in hMDMs and freshly isolated KCs

**Fig. 6 | DPI stimulates the expression of glycolytic genes but suppresses the expression of inflammatory genes in KCs from HFD-fed mice. a** Comparison of gene expression in KCs isolated from mice fed with NC or HFD. Single-cell suspensions were prepared from mice from Fig. 5a after 8 weeks on HFD (6 mice per group), stained with anti-F4/80, anti-CD11b, and anti-Gr-1. F4/80[+]CD11b[+]Gr1[low] macrophages were purified by cell sorting followed by RNAseq. Shown are differentially expressed genes among the three groups. **b** Functional enrichment analysis of differentially expressed genes based on comparison of KCs from HFD-fed and NC-fed mice or from HFD-fed mice treated with DPI or vehicle. **c** GSEA of gene expression profiles of KCs either from HFD and NC mice or from HFD mice treated with DPI or vehicle. Orange and blue graphs in b and c indicate up- and downregulated pathways as labeled. **d** Macrophage polarization index analysis based on the expression profile in (**a**) above using the online software MacSpectrum (https://macspectrum.uconn.edu)[40]. M1-type polarization is expressed as positive scores, whereas M2-type polarization is expressed as negative scores. **e, f** Purified F4/80[+] liver macrophages from DPI-treated or vehicle-treated mouse livers were incubated with or without 100 ng/mL LPS for 6 h. Cytokines TNF-α, IL-1β, and IL-6 in the culture supernatant were quantified by ELISA (**e**) and intracellular ROS was assayed by CM-H2DCFDA staining (**f**). Each dot in **e** represents one mouse ($n = 5$) from one experiment. Data are presented as the mean ± sd. Representative histograms of ROS levels in **f** were shown. $P$ values were calculated by the two-sided student's $t$-test. * $P < 0.05$, ** $P < 0.01$, *** $P < 0.001$, **** $P < 0.0001$. Source data are provided as a Source Data file.

from B6 mice in vitro or systematically in normal B6 mice (Supplementary Fig. 11a–c), Similarly, DPI treatment of HFD-fed mice significantly inhibited production of ROS and proinflammatory cytokines TNF-α, IL-1β, and IL-6 in KCs under LPS stimulation (Fig. 6e, f). In general, inflammatory macrophages, as well as activated T cells, use glycolysis for energy metabolism, and glycolysis orchestrates inflammation but is not the cause of inflammation. Our findings that DPI stimulates glycolysis without stimulating inflammatory cytokine production suggest that the two processes are not mechanistically linked but separable. Consistent with this interpretation, DPI does not upregulate the gene modules that are upregulated by IFNγ or LPS[15] but suppresses LPS-stimulated production of ROS at both transcriptional and protein levels (Fig. 6 and Supplementary Fig. 11b). Furthermore, the pentose phosphate pathway (PPP) and ROS are required for inflammatory cytokine production[44,45]. However, DPI does not upregulate the expression of G6PD, which is a key enzyme for switching to PPP[15]. Thus, DPI stimulates macrophage glycolysis without inducing inflammation, leading to reduced liver pathologies in HFD-fed mice.

It is now generally accepted that metabolism is the core process underlying all biological phenomena, providing energy and building blocks for macromolecules[46]. In particular, inflammatory macrophages and T helper cells 1 (Th1) and 17 (Th17) are known to use glycolysis as the main source of energy production and glycolytic metabolites as part of the inflammatory responses. Studies have shown that limiting glycolysis blocks cytokine production but not proliferation[47], suggesting a critical role of glycolysis in orchestrating inflammatory response. Our findings that DPI stimulates glycolysis without inducing inflammatory cytokine and ROS production suggest that the two processes are not coupled mechanistically but separable. While how the two processes are dissociated in macrophages following DPI treatment remains to be elucidated, our findings that GPR3 activation, as shown by DPI, could be particularly suited for activating macrophage function in chronic diseases without the possible side effects due to the production of cytokines and ROS.

What is even more remarkable is our observation that knockout of PKM2 specifically in KCs abolishes the effect of DPI on HFD-induced obesity and liver pathogenesis in mice (Fig. 5 and Supplementary Fig. 9). At the physiological level, DPI failed to inhibit HFD-induced weight gain, reduce the levels of TAG and lipid droplet deposition in the liver, serum AST and ALT, and blood insulin and response to glucose challenge in KC-specific $Pkm2^{-/-}$ mice as compared to control $Pkm2^{f/f}$ mice. At the biochemical level, DPI (at 50 nM) failed to stimulate ECAR, glycolytic capacity, and glycolytic reserve in $Pkm2^{-/-}$ KCs as compared to $Pkm2^{f/f}$ KCs (Supplementary Fig. 10). Furthermore, DPI did not stimulate glucose consumption, increase in glycolytic intermediates and decrease in TCA cycle intermediates in $Pkm2^{-/-}$ BMDMs as compared to wildtype BMDMs (Fig. 3b, c and Supplementary Fig. 4a–c). As $Pkm2^{-/-}$ BMDMs and $Pkm2^{-/-}$ KCs behaved the same way in glycolysis and bioenergetic analyses in response to DPI, $Pkm2^{-/-}$ KCs likely have similar alterations in metabolites as $Pkm2^{-/-}$ BMDMs. In addition, DPI at 50 nM did not increase the ECAR of hepatocytes or adipocytes (Supplementary Fig. 8), suggesting that the effect of DPI on

HFD-induced obesity and liver pathogenesis is not through elevation of glycolysis in these cell types in mice. Together, these results suggest a key role of PKM2 expression in KCs in mediating the effect of DPI at the organismal level.

How does PKM2 shape the metabolism of KCs? We noticed that among pyruvate kinase isoforms, $PKM2$ was much more highly expressed than $PKM1$ in KCs, and the level of $PKL$ was at the background level (Supplementary Fig. 7). In addition, DPI treatment upregulated PKM2 expression but not PKM1. Thus, one way that PKM2 shapes the metabolism of KCs is because PKM2 is the dominant isoform of pyruvate kinase expressed in KCs. In this respect, one possible outcome is whether $Pkm2^{-/-}$ KCs have functional metabolic pathways. The following observations suggest that $Pkm2^{-/-}$ KCs have functional metabolic pathways. Firstly, $Pkm2^{-/-}$ KCs responded to glucose, oligomycin, and 2-DG, although they have a lower basal level of ECAR (Supplementary Fig. 10), suggesting that they are metabolically active probably because of compensation by PKM1 or PKM2 is not essential for maintaining the minimal level of basal glycolytic activity. Secondly, we were able to isolate similar numbers of KCs from KC-specific $Pkm2^{-/-}$ mice and control $Pkm2^{f/f}$ mice and there was no noticeable difference in their survival following culture in vitro for 30 hours, suggesting that Pkm2-deficiency does not compromise KC differentiation and survival in mice. Similarly, we did not notice any difference in differentiation, survival, and growth between $Pkm2^{-/-}$ BMDMs, $Gpr3^{-/-}$ BMDMs, $Gpr3^{-/-}$ ImKCs, and $Arrb^{-/-}$ ImKCs and their wildtype counterparts. Finally, we did not observe any growth defect under normal diet between KC-specific $Pkm2^{-/-}$ mice and $Pkm2^{f/f}$ mice, or even between germline $Pkm2^{-/-}$ mice and $Pkm2^{+/+}$ control mice[48]. These observations support that PKM2-deficient KCs have functional metabolic pathways.

Our in vitro and in vivo findings raise an intriguing question as to how KCs exert such a huge effect at the organismal level. One possibility is that in addition to its direct role in metabolism and inflammation, KCs could function as a master regulator to control metabolic activities and energy metabolism of other cell types, such as adipocytes and hepatocytes[4,6,7]. While the detailed molecular mechanism remains to be elucidated, our findings suggest that metabolic reprogramming of KCs alone is sufficient to protect mice from HFD-induced obesity and liver pathogenesis.

Finally, our scRNAseq analysis shows the presence of DAMs in the liver of NAFLD patients, which share the same phenotype, including expression of TREM2, CD9, GPNMB, MHCII (HLA-DRB1), C1QA, and CLEC10A, as those found in the livers of patients with NASH and cirrhosis. At the transcription level, the DAMs are inhibited in glycolysis but increased in inflammation, as suggested by the downregulation of glycolytic genes and upregulation of inflammatory genes. Importantly, KCs, including DAMs from NAFLD patients, respond to DPI by upregulating the transcription of glycolytic genes and downregulating the transcription of inflammatory genes, suggesting DPI is also effective at reprogramming the metabolism of human KCs.

In summary, our study identifies a new pathway involving activation of GPR3, β-arrestin2, PKM2, and c-Myc in switching cellular

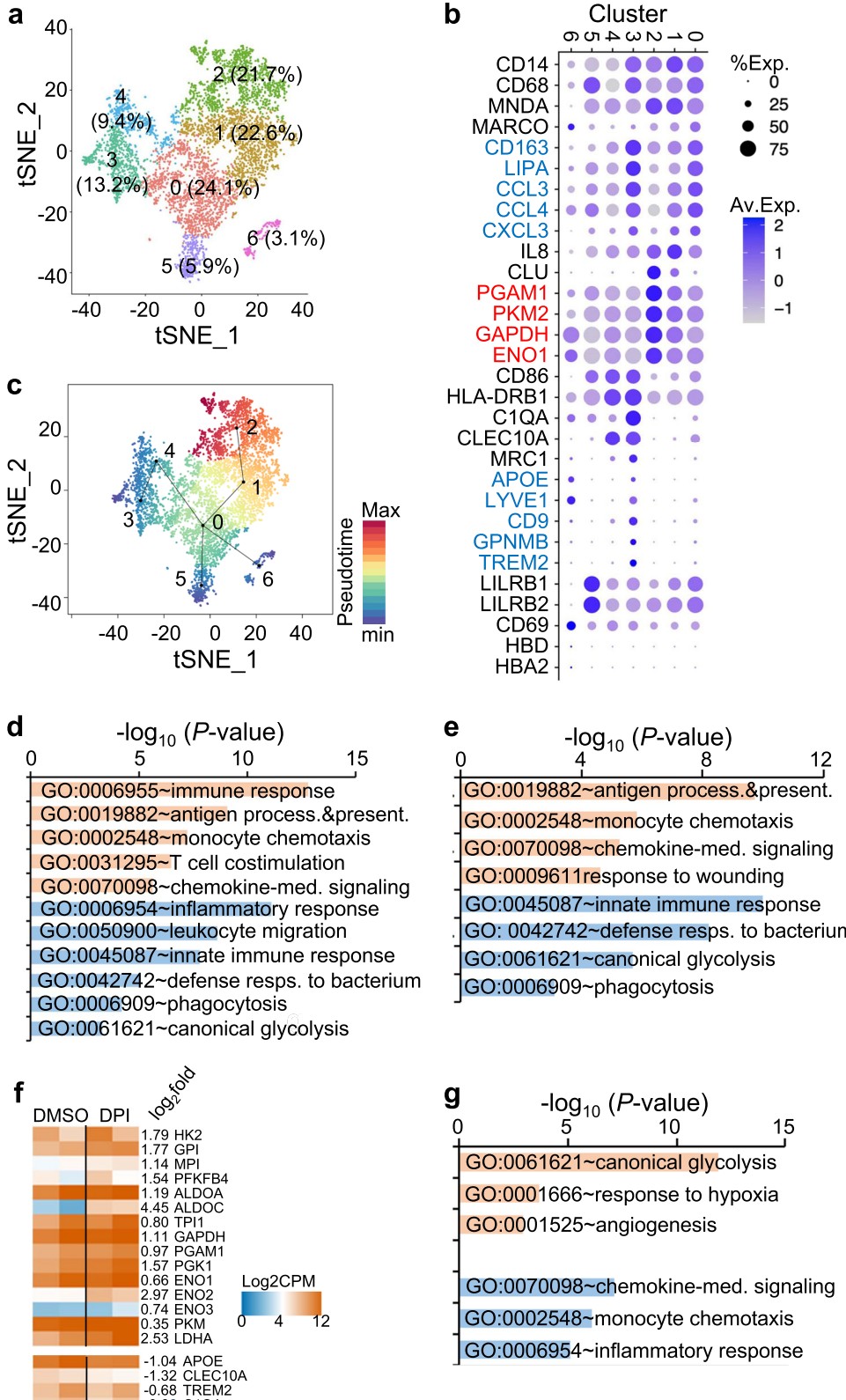

metabolism from oxygen consumption to glycolysis in macrophages, a critical role of KCs in protection from HFD-induced obesity and liver pathogenesis, and the potential of targeting this pathway in DAMs for therapeutic interventions in obesity and NAFLD. Furthermore, since DAMs with similar phenotypes have been observed in various tissues with diverse pathologies, such as HFD-induced NASH in mice[13], scar tissues[12], Alzheimer's disease[49,50], and lung fibrosis[51,52], reprogramming macrophage metabolism, such as by DPI, could be a general therapeutic approach to treat these diverse diseases.

## Methods

### Ethics statement

All uses of human PBMC in this research have been approved by the Institutional Review Board at the Massachusetts Institute of

**Fig. 7 | DPI upregulates glycolysis and suppresses inflammatory responses of Kupffer cells from patients with NAFLD. a, b** scRNAseq analysis of the liver macrophage populations. A total of 5497 macrophages based on the expression of CD14 and CD68 (clusters 5, 8, and 12 in Supplementary Fig. 10c) were subjected to clustering analysis by tSNE. A total of 7 clusters were identified (**a**). Each cluster was annotated based on the expression of typical markers or specific functional markers, as shown by dot plot (**b**). **c** Trajectory inference of the liver macrophages by slingshot[60]. **d, e** GO enrichment analysis of differentially expressed genes (DEGs) between C3 and C0 (**d**) or C3 and C1/2 (**e**). **f, g** Comparison of gene expression

changes induced by DPI in primary KCs isolated from NAFLD liver biopsies. CD14+ KCs were sorted from single cell suspensions of NAFLD human liver biopsies (*n* = 2) and treated with DMSO or DPI (500 nM) for 24 h, followed by RNAseq to quantify gene expression. Shown are the expression changes of glycolytic genes and DAM markers (**f**) and GO enrichment analysis of DEGs induced by DPI in KCs (**g**). Orange and blue graphs in **d**, **e**, and **g** indicate upregulated and downregulated pathways as labeled. *P* values in **d**, **e**, and **g** were computed by Fisher's Exact test. Source data are provided as a Source Data file.

Technology (MIT). All animal studies and procedures were approved by the MIT Committee for Animal Care. Human liver biopsies were collected with the approval of The Ethics Committee of The First Hospital of Jilin University (19K098-001). Written informed consent was obtained from the legal representative of each donor.

### Mice, antibodies, cell lines, and plasmids

C57BL/6 (B6) mice, *p47phox*[-/-], and Clec4f-Cre mice were purchased from the Jackson Laboratory, and *Pkm2*[fl/fl] mice were described in the previous publication[32]. All animals were maintained in the animal facility at the Massachusetts Institute of Technology (MIT) in standard plastic cages under controlled humidity (50%) and temperature (23 ± 1 °C) conditions in a 12 h/12 h dark/light cycle. Mice will be humanely sacrificed with $CO_2$ narcosis painlessly and as quickly as possible using the equipment in the institutional facility. Bone marrow from *Gpr3*[-/-] mice was a kind gift from Dr. Grzegorz Godlewski (NIAAA/NIH). Antibodies specific for CD11b (M1/70), F4/80 (BM8), CD45.2 (104), and Gr-1 (RB6-8C5) for flow cytometry were from BioLegend. Anti-PKM2 (1C11C7, #60268) was purchased from Thermo Fisher Scientific. Anti-GPR3 (#sc390276) was from Santa Cruz Biotechnology. Anti-β-arrestin2 (#3857), Glycolysis Antibody Sampler Kit (#8337 and #12866), anti-Lamin B1 (#12586), anti-β-tubulin (#56739) or anti-β-actin (#5125) were from Cell Signaling Technology. β-Arrestin2 CRISPR plasmids (#sc432139) were from Santa Cruz Biotechnology. pCMV-β-arrestin2-GFP, pCMV-β-arrestin1-GFP and pCMV6-Flag-Myc-β-arrestin2 were from Origene. Immortalized Kupffer cell line (ABI-TC192D, AcceGen), human primary KCs (ABC-TC4369, AcceGen), THP-1 (ATCC TIB-202), and 293 T (CRL-3216) were cultured following vendor instructions (37 °C, 5% $CO_2$). CRISPR/Cas9 system was used to knockout *Gpr3* or *Arrb2* in ImKCs with two *Gpr3* gRNAs: CACCAC-CAGCGCGTTCTCGC and AACATGGGGGCGCGGAAGGC, or two *Arrb2* gRNAs: TCTGTTCATCGCCACCTACC and TGTGGATCCTGAC-TACTTGA. gRNAs were cloned into Lenti-crispr v2 plasmid and lentivirus were produced in 293 T cells with psPAX2 pMD2.G plasmids using TransIT@-Lenti (Mirus). Knockout clones were confirmed and validated by qPCR and Western blotting. Transfection of ImKCs with siRNAs was accomplished using Lipofectamine™ 2000 (Thermo Fisher Scientific) according to the manufacturer's instructions. NOX inhibitor apocynin and ERK1/2 inhibitor SCH772984 were purchased from APExBIO.

### Mouse bone marrow-derived macrophages (BMDMs) and human monocyte-derived macrophages (hMDMs)

Mouse BMDMs were prepared as described previously[15]. Briefly, fresh bone marrow cells were isolated from B6 mice, plated onto a six-well plate with 1 × 10⁶/mL in complete RPMI with 2-mercaptoethanol and with 50 ng/mL M-CSF (PeproTech) for 6 days with fresh medium change every 2 days. Human monocytes were purified from the PBMC of healthy donors (Research Blood Components LLC, USA) and cultured in complete RPMI in the presence of M-CSF (Peprotech, #300-25) for 7 days. hMDMs were treated with 50 or 500 nM DPI for 24 h. For phagocytosis assay, hMDMs (2 × 10⁴ cells) were incubated with 15 µg/mL pHrodo Green *E. coli* BioParticle (ThermoFisher, #P35366) at 37 °C for 1 h, and phagocytosis was quantified by flow cytometry. For assaying ROS production, hMDMs were incubated with 5 µM CM-

H2DCFDA (ThermoFisher, #C6827) at 37 °C for 20 min, and total ROS were quantified by flow cytometry.

### Cytokine arrays

hMDMs were treated with DMSO or 50 nM DPI for 24 h. The culture supernatant was collected and an equal amount from four donors was mixed and applied to the human cytokine antibody array (#AB169827, Abcam) according to the manufacturer's manual. The array has a panel of antibodies to detect 60 human cytokines or chemokines with duplicate spots for each on the film. The films were scanned by ImageQuant LAS 4000 (Cytiva) with the same exposure time. To examine whether DPI induces inflammation in vivo, wildtype B6 mice were dosed with 2 mg/kg DPI i.p. and plasma was collected before dosing and 48 h after dosing. An equal amount of plasma from four mice was mixed and applied to the mouse cytokine antibody array (#AB133995, Abcam) according to the manufacturer's manual. The array has a panel of antibodies to detect 62 mouse cytokines or chemokines with duplicate spots for each on the film. The films were scanned by ImageQuant LAS 4000 (Cytiva) with the same exposure time.

### Co-immunoprecipitation, Western blotting, and native PAGE

293 T cells were transfected with FLAG-tagged β-arrestin2, using TransIT®-LT1 Transfection Reagent (Mirus). Forty-eight hours after transfection, the cells were lysed using a cold Lysis Buffer containing 20 mM Tris-HCl (PH 7.4), 150 mM NaCl, 0.1% NP-40, 10% glycerol, proteinase inhibitor (Roche, # 11836153001), and phosphatase inhibitors (Roche, #04906845001). The clear supernatants from the lysate were incubated with M2-magnetic beads conjugated with anti-FLAG antibody (Sigma, #M8823) for 2 h at 4 °C. Then the beads were washed twice and eluted by the 3×FLAG peptides (Sigma, #F4799) as described in the manufacturer's manual for Western blotting.

Proteins were extracted from cells with RIPA buffer. Protein concentration was quantified by the BCA Protein Assay Kit (Pierce Biotechnology). Samples containing 60 µg total protein were resolved on a 10% SDS-PAGE gel and electro-transferred onto a PVDF membrane (Millipore Corporation). The membrane was blocked in 5% (w/v) fat-free milk in PBST (PBS containing 0.1% Tween-20). The blot was hybridized overnight with primary antibodies according to the recommended dilution in 5% fat-free milk. The blot was washed twice in PBST and then incubated with HRP-conjugated secondary antibody (Cell Signaling Technology) in 5% fat-free milk with the dilutions recommended by the manufacturer manuals. The membrane was washed twice in PBST and subjected to protein detection by ECL Plus Western Blotting Detection System (GE Healthcare) before being exposed to a Kodak BioMax XAR film. The membrane was stripped and reblotted with the anti-β-tubulin or anti-β-actin for protein loading control.

To analyze the cytosolic and nuclear proteins, cells were lysed and proteins of nuclear and cytoplasmic fractions were extracted using Nuclear and Cytoplasmic Extraction Reagents (Thermo Fisher Scientific, #78833) according to the manufacturer's protocol. The indicated proteins were analyzed by Western blotting. The loads of cytosolic proteins and nuclear proteins were normalized to the amount of GAPDH and Lamin B1.

To analyze the proteins in the naïve PAGE gel, proteins were extracted from cells in 1× Native PAGE sample buffer (Thermo Fisher

Scientific) containing 1% digitonin followed by 20 min spin at 12,000 $g$ to pellet debris. Protein extracts were analyzed using NativePAGE Novex System (Thermo Fisher Scientific) and subsequently transferred to PVDF membrane, fixed, and blotted for native proteins.

## Metabolite profiling

ImKCs were treated with DPI (#81050, Cayman Chemical) at 50 or 500 nM for 6 h or 24 h. *Pkm2*−/− BMDMs were treated with DPI at 50 or 500 nM for 24 h. Cells were washed once in ice-cold 0.9% NaCl, and lysates were extracted in 80% methanol solution containing internal standards for LC/MS by scraping on dry ice followed by 10-minute mixing with vortex at 4 °C. Following lysate extraction, debris was removed by high-speed centrifugation and supernatants were dried using speed vac. Samples were analyzed by LC/MS on QExactive Orbitrap instruments (Thermo Fisher Scientific) in the Whitehead Institute Metabolite Profiling Core Facility. Data analysis was performed using the in-house software described previously[53]. The raw data was provided in Supplementary Data 4.

To compare metabolites in WT and *Gpr3*−/− ImKCs, WT and *Gpr3*−/− ImKCs were pretreated with DMSO or 50 nM DPI for 24 h. Cells were collected and resuspended in prechilled 80% methanol by vortex. Then, cells were melted on ice and whirled for 30 s. After the sonication for 6 min, the samples were centrifuged at 5000 rpm, 4 °C for 1 min. The supernatant was freeze-dried and dissolved with 10% methanol. The supernatant was analyzed by Novogene Co., Ltd. (Beijing, China) using an ExionLC™ AD system (SCIEX) coupled with a QTRAP ® 6500+ mass spectrometer (SCIEX). The selected metabolites were quantified using Multiple Reaction Monitoring based on Novogene's in-house database. The raw data was provided in Supplementary Data 5.

## β-arrestin membrane translocation assay

BMDMs or ImKCs were transfected with plasmids encoding pCMV-β-arrestin1-GFP or β-arrestin2−GFP. Twenty-four hours after transfection, cells were reseeded into a 24-well glass-bottom plate (MatTek Life Sciences) and treated with DPI (50 nM), S1P (3 mM), or vehicle control (0.3% DMSO) for the indicated duration. The fluorescent signals of β-arrestin2 were measured in live cells using a total internal reflection fluorescence (TIRF) microscope (Olympus).

## Measurements of oxygen consumption rate, extracellular acidification rate, and glucose consumption

Oxygen consumption rate (OCR) and extracellular acidification rate (ECAR) were measured in cultured ImKCs or BMDMs using the Seahorse XF Extracellular Flux Analyzer (Agilent). For the immediate effects of DPI on OCR or ECAR, $2 \times 10^4$ cells per well were seeded into the XF96 cell culture microplate and incubated overnight. Cells were washed and maintained in prewarmed XF assay media supplemented with 10 mM glucose, 1 mM sodium pyruvate, and 2 mM L-glutamine. DMSO, DPI (5, 50, and 500 nM), and 2-deoxyglucose (2-DG, final 50 mM for ECAR) or rotenone/antimycin A (Rot/Aa, final 1 μM for OCR) were added at the indicated times. For other ECAR measurements, $2 \times 10^4$ cells per well were seeded into the XF96 cell culture microplate and treated with DMSO or DPI (50 and 500 nM) for 24 h. Cells were washed and maintained in prewarmed XF assay media supplemented with 1 mM sodium pyruvate and 2 mM L-glutamine. At the indicated time, glucose (final 10 mM), oligomycin (final 2 μM), and 2-DG (final 50 mM) in XF assay media were injected. Data sets were analyzed using XFe96 software (Agilent). Glycolytic capacity is calculated as (maximum rate measured after glucose stimulation) − (maximum rate measured after oligomycin stimulation). The glycolytic reserve is calculated as (maximum rate measured after oligomycin stimulation) − (rate measured before oligomycin stimulation). The glucose consumption was measured by the glucose uptake assay kit (Abcam) according to the manufacturer's protocol. Briefly, cells were treated with DMSO or DPI (50 and 500 nM) for 24 h in the presence of the fluorescent glucose analog 2-NBDG. The mean fluorescence intensity (MFI) of 2-NBDG in cells was quantified by flow cytometry.

## Immunofluorescent staining and microscopy

BMDMs or Kupffer Cells were fixed and incubated with primary antibodies and then labeled with Alexa Fluor dye-conjugated secondary antibodies and counterstained with Hoechst 33342 according to standard protocols. Cells were examined using a deconvolution microscope (Zeiss) with a 63-Å oil immersion objective. Axio Vision software from Zeiss was used to deconvolute Z-series images.

## PKM and GAPDH enzymatic activity assay

The enzymatic activities of PKM and GAPDH were measured using the pyruvate kinase activity assay kit (Biovision, # K709) and GAPDH activity assay kit (Biovision, # K680) according to the manufacturer's protocols, respectively. Briefly, $5 \times 10^5$ cells were homogenized with 100 μl of GAPDH assay buffer. Samples were kept on ice for 10 min and centrifuged at 10,000 $g$, 4 °C for 5 min. The absorbance at 450 nm was measured every 5 or 10 min. The experiments were performed in triplicates with three independent experiments.

## Myc luciferase reporter assay

The c-Myc activity was assessed using the Myc Reporter kit (BPS Biosciences) and the Dual-Luciferase Reporter System (Promega) according to the manufacturer's instructions. Briefly, 100 μL ($1.5 \times 10^5$ cells/mL) of wildtype and *Pkm2*−/− KCs were seeded into 96-well plates. After overnight incubation, when cells reached ~50% confluency, 1 μL of Reporter A (60 ng/μL) was transfected into cells using Turbofectin 8.0. After 48 h, cells were lysed in 25 μL Passive Lysis Buffer (provided in the Dual-Luciferase Reporter kit). 20 μL of cell lysate was transferred to 96-well plates and placed in a 96-well microplate luminometer (GloMax-Multi, Promega). 100 μL Luciferase Assay Reagent II and 100 μL Stop & Glo Reagent (both provided in the Dual-Luciferase Reporter kit) were sequentially injected, and firefly and Renilla luciferase activities were automatically measured. c-Myc activities were determined by the ratios of firefly to Renilla luciferase activities.

## HFD-induced NAFLD mouse model and treatment

C57BL/6 male mice at 5 weeks of age (body weight = 18−20 g) were fed with a normal chow (NC) diet (10 kcal% fat, Research Diets, #12450 J) or high-fat diet (HFD) (60 kcal% fat, Research Diets, #D12492) for a total of 8 weeks to induce obesity. Three weeks after the HFD, mice were randomly assigned into two groups: the HFD + vehicle group was injected with the vehicle (PEG3000) and the HFD + DPI group was injected with DPI in the vehicle (2 mg/kg) i.p. every 5 days for 4 weeks. In the *Pkm2*−/− mouse experiment, Kupffer cell-specific *Pkm2*−/− male mice at 5 weeks of age were fed with HFD for a total of 8 weeks. Three weeks after onset on the HFD diet, the mice were randomly assigned to two groups for treatment with vehicle or DPI as described for B6 mice. To induce liver hepatosteatosis, C57BL/6 male mice at 5 weeks of age were fed with HFD for a total of 16 weeks. After 9 weeks on HFD, the mice were randomly assigned to two groups for treatment with vehicle or DPI (2 mg/kg) i.p. every 5 days for 5 weeks. The weight of the mice and food consumption were monitored weekly. At the end of the experiments, livers were perfused with PBS with 1 mM EDTA and fixed with 10% buffered formalin. Samples were embedded in paraffin, sliced (5 μm sections), and stained with hematoxylin and eosin (H&E) for histopathological analysis.

## Kupffer cells isolation from mouse livers

Mouse livers were perfused with 30 mL of HBSS with 0.5 mM of EDTA and 25 mM of HEPES and then dissected and digested with Collagenase IV (Roche). Single cells were pelleted and resuspended in 450 μl of PBS containing 2% FBS after red blood cells were removed. Kupffer cells

were purified using anti-F4/80 microbeads (Thermo Fisher Scientific, #8802-6863-74) with an LS column (Miltenyi Biotec,130-042-401). The enriched liver macrophages (consisting of 80-90% F4/80$^{hi}$CD11b$^{int}$Gr1$^{low}$ KCs and 10–20% of F4/80$^{int}$CD11b$^{hi}$Gr1$^{low}$ MDMs) were seeded into 6-well plates pre-coated with collagen. After plating for 1-2 hrs, cells were washed with PBS and cultured in a fresh medium overnight for glycolysis analysis, ROS, and cytokine production. Fresh KCs from DPI-treated or vehicle-treated mice were stimulated with or without 100 ng/mL LPS for 6 h and cytokines TNF-α, IL-1β, and IL6 in the supernatant were measured by the TNF-α ELISA kits (Dakewe, # 1217202), the IL-1β ELISA kit (Dakewe, #1210123) and IL-6 ELISA kit (Dakewe, #1210602), respectively, according to the manufacture's manuals. Fresh KCs from B6 mice were first treated with or without DPI at indicated concentrations for 24 h and then stimulated with 100 ng/mL LPS in fresh medium (without DPI) for ROS and cytokine analysis.

### Glucose tolerance test, blood insulin, and liver TAG

The glucose tolerance test was performed in mice at the indicated time after feeding with HFD or NC. Mice were fasted overnight, followed by an intraperitoneal injection of 1 g/kg glucose. Blood was obtained from the tail vein before (0 min) and 15, 30, 60, 90, and 120 min after the injection of glucose. Glucose levels were measured using an automatic glucometer (Bayer Contour Next One meter). Blood was collected from mice into PBS/EDTA before glucose tolerance assay and plasma were harvested. Plasma insulin levels were quantified by the Insulin Enzyme Immunoassay (#589501, Cayman Chemical) according to the manufacturer's manual. Liver tissues were perfused and fixed in 10% formalin. Lipids were extracted from liver tissues using a lipid extraction kit (#ab211044, Abcam) according to the manufacturer's manual. TAG levels were quantified by a colorimetric assay (#10010303, Cayman Chemical) according to the manufacturer's protocol.

### Immune cell and Kupffer cell purification from human liver biopsies

Human liver biopsies were from livers from deceased donors procured for liver transplantation. In the case of healthy livers and livers with mild NAFLD, biopsies were from leftover liver tissues used for pathological diagnosis. In the case of livers with moderate to severe NAFLD, biopsies were from livers deemed unacceptable for liver transplantation. All experiments were performed in accordance with the relevant guidelines and regulations. During organ retrieval, donor liver grafts were perfused in situ with cold HTK solution (Methapharm) to thoroughly flush out circulating cells. The tissue was gently dissected with a sterile scissor into 1 mm$^3$ slices on ice. The slice was incubated in D-HBSS containing 100 U/mL collagenase II (Giboco, #17101015) and 10 U/ml DNase I (Thermo Scientific, #89836) and shaken at 37 °C for 1.5 h. Single cells were stained with anti-CD45 (#30406, Biolegend) to purify all immune cells for scRNAseq or with anti-CD45 and anti-CD14 (#301804, Biolegend) to purify liver macrophages (the majority of which are considered as KCs) for in vitro treatment by flow cytometry (BD Aria).

### RNA isolation, sequencing, and data analysis

Mouse livers were dissected and digested with Collagenase IV (Roche). Single-cell suspensions were stained with anti-F4/80, anti-CD11b and anti-Gr-1. F4/80$^+$CD11b$^+$Gr1$^{low}$ macrophages (consisting of 80–90% KCs) were sorted by flow cytometry (BD Aria). RNAs were extracted with Rneasy MinElute Kit (Qiagen) from sorted mouse liver macrophages, converted into cDNA, and sequenced using Next-Generation Sequencing (Illumina). RNA-seq data were aligned to the mouse genome (version mm10) or human genome (version hg19) and raw counts of each gene of each sample were calculated with bowtie2 2.2.3[54] and RSEM 1.2.15[55]. Differential expression analysis was performed using the program edgeR at $P < 0.05$ with a twofold change[56]. The gene expression level across different samples was normalized and quantified using the function of cpm. Differentially expressed genes (DEGs) were annotated using the online functional enrichment analysis tool DAVID[57]. To quantify the levels of RNA transcripts, total RNA was extracted from various cells and reverse transcribed by TaqManTM Reverse Transcription Reagents Kit (Applied Biosystems, #N8080234), followed by amplification with SYBR Green Master Mix (Roche, #04707516001) by Roche LightCycler 480 with specific mouse primers: *Pkm2*: 5′ primer-TCGCATGCAGCACCTGATT, 3′ primer-CCTCGAATAGCTGCAAGTGGTA; *Pkm1*: 5′ primer-GTCTGGAGAAACAGCCAAGG, 3′ primer-TCTTCAAACAGCAGACGGTG; *Ldha*: 5′ primer-GCAGACAAGGAGCAGTGGAAGGAG, 3′ primer-ACACTGAGGAAGACATCCTCATTG; *Hk2*: 5′ primer-CCTGCTACAGGTCCGAGCCATCTT, 3′ primer-GAGGATGAAGCTTGTACAGTGTCC; *Myc*: 5′ primer-ATGCCCCTCAACGTGAACTTC, 3′ primer-GTCGCAGATGAAATAGGGCTG; *β-actin*: 5′ primer-GTGACGTTGACATCCGTAAAGA, 3′ primer-GCCGGACTCATCGTACTCC. The Ct values were normalized with the housekeeping gene *β-actin* comparison.

### Single-cell RNAseq and computational analysis

Sorted human CD45$^+$ cells were resuspended and washed in 0.05% Rnase-free BSA in PBS for single-cell library preparation with 10× Chromium Next GEM Single Cell 3′ Kit (10× genomics) according to the manufacturer's instructions. The single-cell cDNA libraries were sequenced by NexSeq500 (Illumina). Raw sequences were demultiplexed, aligned, filtered, barcode counting, and unique molecular identifier (UMI) counting with Cell Ranger software v3.1 (10× genomics) to digitalize the expression of each gene for each cell. The analysis was performed using the Seurat 3.0 package[58]. We first processed each individual data set separately prior to combining data from multiple samples. The outlier cells with low number (<500) or high number (>5000) of gene features as doublets, or low total UMI (<1000) and high mitochondrial ratio (>15%) from each data set were removed. Subsequently, samples were combined based on the identified anchors for the following integrated analysis. We ran principal component analysis (PCA) and used the first 15 principal components (PCs) based on the resampling test with the JackStraw procedure to perform tSNE clustering. We checked well-defined marker genes for each cluster to identify potential cell populations, such as T cells (CD3E, CD8A, CD4, CD69, IL7R), B and plasma cells (CD19, *MS4A1, SDC1*), dendritic cells (CD11C, CLEC9A), natural killer cells (CD56, CD16, GZMB). For macrophage analysis, CD14 and CD68 positive clusters were selected for subsequent analyses. We repeated PCA tSNE clustering on the integrated data sets of macrophages. Differential expression analysis was performed in each cluster compared with other indicated cells by FindMarkers function with setting min.fct = 0.25 and logfc.threshold = 0.25. Differential expression analysis was performed in each cluster between NAFLD and healthy donors with setting min.fct = 0.25 and logfc.threshold = 0.1. For gene sets representing specific cellular functions or pathways, we performed functional enrichment analysis with the biological process of Gene Ontology by the online tool DAVID[57] or GSEA[59] with hallmarker gene-sets by GSEAPreranked procedure with the ranked list of genes according to folder change of expression. Trajectory inference of the liver macrophages was analyzed by slingshot using the reduced dimensions of tSNE generated by Seurat analysis.

### Statistic methods

Statistical significance was determined with the two-sided unpaired or paired Student's $t$-test. The FDRs were computed with $q = P \times n/i$, where $P = P$ value, $n =$ total number of tests, and $i =$ sorted rank of $P$ value.

### Reporting summary

Further information on research design is available in the Nature Portfolio Reporting Summary linked to this article.

## Data availability

Raw RNA-seq data are deposited in the database of Gene Expression Omnibus with accession IDs: GSE186329. Metabolic raw data were deposited in OpenScienceFramework (OSF) (https://doi.org/10.17605/OSF.IO/EW5YF). The metabolic raw data and DEGs are also provided as Supplementary Data. All other data that support the findings of this study can be found in the manuscript, figures, and supplementary data or are available from the corresponding authors upon request. Source data are provided with this paper as a source data file. Source data are provided in this paper.

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

## Acknowledgements

The authors would like to thank Koch Institute Swanson Biotechnology Center and core facilities for assistance with the flow cytometry, histology, and RNA-seq data acquisition and analysis, and the Translation Medicine Core Facility in Shandong University for assistance in metabolite profiling of ImKCs. The authors would like to thank members of Chen Lab for their suggestions. This work was supported in part by National Institutes of Health Grants CA197605 and NS104315 to J.C., ESO28374 to M.B.Y., Ivan R. Cottrell Professorship and Research Fund, the Koch Institute Support (core) Grant P30-CA14051 from the National Cancer Institute, the MIT Center for Precision Cancer Medicine, and the National Research Foundation of Singapore through the Singapore–MIT Alliance for Research and Technology's Interdisciplinary Research Group in Antimicrobial Resistance Research Program.

## Author contributions

G.H., T.D., and J.C. designed the research, interpreted the data, and wrote the paper. T.D., G.H., Z.F., S.W., H.X. H.W., and Y.G. performed experiments. G.H., T.D., M.V.H., and J.C. analyzed the data. M.Y. provided mice and edited the paper. G.L. contributed to the discussion and obtained human liver biopsies.

## Competing interests

G.H., T.D., and J.C. (inventors) declare that a provisional patent application related to this work has been filed with the United States Patent and Trademark Office on July 21, 2021. The other authors declare no competing interest, but M.G.V.H. discloses that he is a scientific advisor for Agios Pharmaceuticals, iTeos Therapeutics, Sage Therapeutics, Faeth Therapeutics, and Auron Therapeutics.
