## [Peer Review File · Nature Communications]

Activation of GPR3- β -Arrestin2-PKM2
pathway in Kupffer cells stimulates glycolysis and inhibits
obesity and liver pathogenesisREVIEWER COMMENTS

Reviewer #1 (Remarks to the Author):

This is an interesting and substantial set of studies that conclude that the molecule diphenyleiodonium (DPI) is able to regulate HFD-induced obesity and liver pathogenesis in mice by enhancing glycolysis and suppressing the inflammatory response of Kupffer cells. The evidence to support this is strong but the evidence that this is produced through activation of the G protein-coupled receptor GPR3 is weak and indirect. Simple ways to assess this, for example the use of GPR3 knock-out mice, are available and it is surprising that these are not used as other knock-out lines to support aspects of the potential downstream mechanism(s) are employed. In addition, although available pharmacological tools to study GPR3 are quite limited, the data to support that effects of DPI are actually mediated via GPR3 could and should be more extensive. These are major limitations in ascribing the observed effects to GPR3 and require further study.

Points

1. Although studies indicating that DPI can cause activation of GPR3 go back to 2014 (J Pharmacol Exp Ther 49(3):437-43), diphenyleiodonium is a very simple small molecule. Early studies did explore whether it was selective, but only a trivial set of GPCRs was explored. Thus we know little to nothing about its overall selectivity. It is potentially interesting that DPI is quite similar in structure to 3,3'-diindolylmethane, an allosteric activator of GPR84, a pro-inflammatory receptor expressed by a wide range of immune cells. Clearer understanding of the target(s) for DPI in the in vivo studies would greatly enhance these studies.
2. In one strand of the work the authors employ cells from p47phox^{-/-} mice to try to eliminate a potential non-GPR3-mediated effect of DPI. However, surprisingly, they do not use GPR3^{-/-} mice although such animals were first described a number of years ago (e.g. Sci Rep 2015 Oct 12;5:14953 and Sci Transl Med 2015 Oct 14;7(309):309ra164). The authors need to explain why they did not use such animals.
3. Even if there is a valid reason to not use GPR3^{-/-} mice then the studies with DPI need to

be augmented with the concurrent use of a GPR3 inverse agonist. In many ways these are as limited as the agonist ligands but molecules such as AF64394 (Bioorg Med Chem Lett. 2014 Nov 15;24(22):5195-8, and J Biomol Struct Dyn. 2021 Jun 23:1-10. doi: 10.1080/07391102.2021) would seem to offer an opportunity to provide greater confidence in the identification of GPR3 as the relevant molecular target.

4 The studies on the immortalised Kupffer cells (ImKCs) are interesting. However, once again in terms of defining the role of GPR3 why is GPR3 'knocked-down' in these cells whilst beta-arrestin 2 is 'knocked-out'? If the gene-editing protocols work effectively in these cells surely it would be most effective to eliminate GPR3 expression?

5. In a small extension to this point why specifically eliminate beta-arrestin 2? Is beta-arrestin 1 also expressed by the cells? These isoforms certainly have overlapping functions and in other cell systems (e.g. HEK293 cells) researchers have selected to knock-out both isoforms and then potentially reconstitute function by selective re-introduction of either isoform.

Minor Point

In supplementary Figure 2 panel c it would be important to show the full immunoblot corresponding to GPR3. Most GPCRs do not migrate as a tight single band in SDS-PAGE due to differential N-glycosylation. The appearance of such a single band is thus rather concerning in terms of potential specificity of the antiserum. No molecular mass markers are provided for these immunoblots and are lacking in a number of other cases.

Reviewer #2 (Remarks to the Author):

In this manuscript entitled ' Activation of GPR3- β -Arrestin2-PKM2 pathway in Kupffer cells protects against high-fat diet induced obesity and liver pathogenesis through enhanced glycolysis' the authors study metabolic regulation of hepatic macrophages during HFD-induced obesity.

By activating the glycolytic rate in macrophages, the authors demonstrate a protection against the harmful metabolic effects of obesity and NAFLD. The authors have used an

extensive approach by including many different experimental setups ranging from in vitro to in vivo animal and human studies.

I feel that the current manuscript may lack some balance with a lot of focus on the mechanistic pathways leading to activation of glycolysis by DPI. Although these results are very important, most likely other pathways would be involved in explaining the in vivo observations with a reduction in BW and improvement in glucose control upon DPI treatment. Although the absence of PKM2 seems to abolish most of the effects of DPI treatment in vivo, a potential underlying mechanisms to explain the metabolic effects is currently lacking. This is even more important since an increase in glycolytic rate and a strong reduction in oxidative phosphorylation after DPI treatment resembles the Warburg effect and is linked to pro-inflammatory activation of macrophages. However, the results in the manuscript show opposite effects of DPI treatment on inflammation. This is hugely interesting, however, it requires careful interpretation and needs to be discussed in light of the current literature that points to glycolysis as a driver of inflammation.

I do have several other comments and suggestion that may help to further improve the quantify of the current manuscript.

1. Figure 1A has not much added value. The metabolic effects of DPI treatment, especially at the highest concentration, are very strong. It seems to almost completely stop oxygen consumption. Any data available on cell viability?

2. Have the authors looked at the effects of LPS stimulation and DPI treatment? LPS is known to rely on an increase in glycolysis to induce a pro-inflammatory response. In addition to studying the mechanism of action in figure 1-4, I would be very interested in functional readouts of the cells. How does this treatment impact on the functional properties of the macrophage (migration, ROS production, cytokine production, phagocytosis) ? This would be crucial to understand the in vivo effects as well.

3. The animal studies are interesting, yet lack several crucial readouts. The authors should quantify TG content in the liver. Other metabolic parameters should be measured including insulin. I am surprised by the data in figure 5c. The HFD seems not to increase eWAT while the increase in total BW is very robust after HFD-feeding. Also, labelling of the different groups is incorrect in figure 5c.

For the in vivo treatment with DPI, I would also be very interested in effects in other tissues. By using IP injections, the compound most likely would affect many other cells including

macrophages residing in different tissues. Keeping in mind the very robust effects on glycolysis presented in the first part of the manuscript, other cells/tissue most likely would be affected by the treatment. It would be important to include effects of the treatment in other cells/tissues as well. The in vivo effects can not be explained by an effect on liver macrophages alone. Although effects on BW/liver TGs are abolished in the knockout model (figure 5g/h), other metabolic parameters are not presented in the manuscript. These data are needed to be able to fully interpret effects of the in vivo treatment with DPI.

It seems the authors are unable to present a mechanism to explain the in vivo observations. Also, a critical reflection using existing studies on the effects of glycolysis on macrophage activation, is lacking.

Reviewer #3 (Remarks to the Author):

Dong et al. studied the effects of DPI on metabolic reprogramming in both mice models and human patients. They showed the activation of GPR4- β -Arrestin2-PKM2 pathway in Kupffer cells could prevent HFD-induced obesity and liver pathogenesis. In addition, they identified a KC population related to disease (DAM) by using single-cell RNA-seq and demonstrated the reversion of glycolysis and inflammation associated aberrations after DPI treatment in vitro. However, the bioinformatics data analysis needs further concerns.

Major

1. In L336, the authors said "C3 was the only elevated LM population" which may refer to Fig. 7b. However, the elevation is largely contributed by NAFLD1, and there is no evident increase of C3 cells in other two patients. In addition, the display of cellular composition alteration in Fig. 7b is not appropriate, which is highly sensitive to the number of cells harvested in each patient. Instead, it is better to calculate the ratio of observed to expected number of cells in each patient and test the significance of C3 increase as compared to healthy donors.
2. Are there any transcriptional alterations in KCs other than C3? In addition to single-cell RNA-seq, this work has several bulk data sets which support most of the clues about gene

expression changes related to glycolysis and inflammation. However, C3 cells are a small portion of KCs in patients according to scRNA-seq. Therefore, it is necessary to compare the transcriptomes of other KCs in patients to healthy donors.

3. Though this study is mainly about macrophages, it is also interesting to investigate the whole immune environment by in-depth analysis of other immune cells in patients as the authors performed single-cell RNA-seq.

Minor

1. L918, "expression profile in a with the", a typo error or missing words after "in a".
2. Fig. 7d is not informative as it is presented this way. A combination of Fig. 7c and d should be better. Otherwise, Fig. 7d should be removed or moved to the sup figure to just indicate a unique expression in each cell cluster.
3. Fig. 7e, cells in C5 and C6 are not shown, is it intended or mistake? It is not clear about the meanings of colors, do colors indicate pseudotime? Details should be added to the legend.
4. Sup. Fig. 9, figure legend, the description "relative proportion of each cluster in each sample (f)" is not correct as (f) displays the composition of donor-derived cells in each cluster instead of in each sample. Again, this type of display is not appropriate.
5. More details about the single-cell RNA-seq data processing are needed in the method section. For instances, batch correction, the rationale of 15 PCs selection, the threshold of determining differentially expressed genes, procedures of performing trajectory inference.
6. The gene lists used to perform GSEA should be summarized to a sup table providing the details of references where the genes are collected.

Reviewer #4 (Remarks to the Author):

The manuscript of Dong et al is an extensive analysis on the role of DPI on the metabolic reprogramming of macrophages and its underlying molecular mechanism. The work is quite comprehensive, however, there are deficiencies in detecting metabolic flux and PKM2 status. The manuscript is not suitable for publication until these issues are resolved.

Introduction

"GPR3 is highly expressed in the brain...neurological processes" The authors did not study the function of GPR3 in the nervous system. The authors should focus here on the research progress of DPI, GPR3 or β -arrestin2 in NFLAD and Kupffer cells, especially on metabolic regulation.

Results

Figure 1: The authors confirmed that DPI promotes the conversion of TCA cycle to glycolysis in glucose metabolism, which is manifested as an increase in ECAR value, an increase in glycolytic metabolites, and a decrease in TCA metabolites. They should first consider whether DPI affects glucose uptake from the extracellular environment. They should identify the difference in metabolite content between DPI-treated cells and control cells under the same amount of glucose uptake (Fig.1f).

Figure 2: Fig.2c is not mentioned in the text.

Figure 3: The authors confirmed that DPI promotes the interaction of PKM2 and β -arrestin2, thereby enhancing glycolysis. Does β -arrestin2 interact with PKM1? We know PKM1 and PKM2 play opposite roles in glucose metabolism. The PKM1 and PKM2 proteins are derived from the same precursor PKM mRNA and differ only by 22 amino acids. Although PKM2 is the main pyruvate kinase subtype in immune cells, tumor cells or embryonic stem cells, the precursor PKM mRNA tends to form PKM1 under some stimulation. They should consider whether DPI affects the alternative splicing of PKM mRNA.

Figure 4: Under DPI treatment, the monomer, dimer and tetramer of PKM2 all appeared to increase, compared with DMSO controlled cells (Fig.4b). Does this mean that DPI directly promotes the expression of the entire PKM2 protein? Due to the high expression of PKM2 caused by DPI treatment, the authors observed an increase in PKM2 bound to β -arrestin2 protein and an increase in the formation of PKM2 dimers. Therefore, the western blotting data cannot support the authors' conclusion that DPI induces the formation of the dimer PKM2.

Figure 4: The authors confirmed the translocation of PKM2 from the cytoplasm to the nucleus by immunofluorescence (Fig.4c). It is also necessary to isolate the nuclear and cytoplasmic components of the KC cells, and detect the expression of nuclear PKM2 and cytoplasmic PKM2 by western blotting.

REVIEWER COMMENTS

Reviewer #1 (Remarks to the Author):

This is an interesting and substantial set of studies that conclude that the molecule diphenyleiodonium (DPI) is able to regulate HFD-induced obesity and liver pathogenesis in mice by enhancing glycolysis and suppressing the inflammatory response of Kupffer cells. The evidence to support this is strong but the evidence that this is produced through activation of the G protein-coupled receptor GPR3 is weak and indirect. Simple ways to assess this, for example the use of GPR3 knock-out mice, are available and it is surprising that these are not used as other knock-out lines to support aspects of the potential downstream mechanism(s) are employed. In addition, although available pharmacological tools to study GPR3 are quite limited, the data to support that effects of DPI are actually mediated via GPR3 could and should be more extensive. These are major limitations in ascribing the observed effects to GPR3 and require further study.

Response: We thank the reviewer for the constructive comments and have revised the manuscript accordingly to address the following comments. To address the reviewer's concern, we have tried to obtain *Gpr3*^{-/-} mice. However, due to pandemic, we were only able to obtain bone marrow cells from *Gpr3*^{-/-} mice from Dr. Grzegorz Godlewski (NIAAA/NIH). We generated bone marrow-derived macrophages (BMDMs) and assayed ECAR and glucose uptake following DPI treatment. DPI at 50 nM failed to stimulate ECAR and glucose uptake by BMDMs (Fig. 2d-e and Supplementary Fig. 2d). We have also constructed *Gpr3*^{-/-} ImKCs by CRISPR-cas9-mediated gene editing. Similarly, DPI (at 50 nM) failed to increase ECAR and glucose uptake by *Gpr3*^{-/-} ImKCs (Supplementary Fig. 2e-j). These results are consistent with the original results using siRNA-mediated knocked down of *Gpr3* in ImKCs. Although 500 nM DPI stimulated ECAR by *Gpr3*^{-/-} BMDMs and ImKCs, the high concentration of DPI also stimulated ECAR in *Arrb2*^{-/-} or *Pkm2*^{-/-} ImKCs (Fig. 2d, 2g-h, 3b-c). In addition, we have provided new data showing that DPI at 50 nM does not stimulate ECAR in hepatocytes and adipocytes (Supplementary Fig. 8), although 500 nM DPI had some effect. Together, these results show that the effect of DPI at 50 nM is primarily through GPR3. The new results have been added into the revised manuscript.

Points

1. Although studies indicating that DPI can cause activation of GPR3 go back to 2014 (J Pharmacol Exp Ther 49(3):437-43), diphenyleiodonium is a very simple small molecule. Early studies did explore whether it was selective, but only a trivial set of GPCRs was explored. Thus we know little to nothing about its overall selectivity. It is potentially interesting that DPI is quite similar in structure to 3,3'-diindolylmethane, an allosteric activator of GPR84, a pro-inflammatory receptor expressed by a wide range of immune cells. Clearer understanding of the target(s) for DPI in the in vivo studies would greatly enhance these studies.

Response: GPR3 shares 60% identity and 80% similarity with GPR6 or GPR12, however only shares 24% identity and 50% similarity with GPR84, which responds to the median-chain-length fatty acids. DPI selectively binds to GPR3 not GPR6/12 (Ye et al. 2014, ref. 26). GPR3 and GPR84 belong to different subclasses of GPCR (alpha and delta) and have different docking pockets based on structure simulations (Capaldi et al. 2018; Mahmud et al. 2017). DPI binds to the orthosteric binding cavity of GPR3 while DIM (3,3'-diindolylmethane) as an allosteric

activator of GPR84 does not appear to bind to the GPR84 orthosteric site. According to these observations, DPI is unlikely to bind to GPR84. Consistent with this interpretation, DPI at 50 nM failed to stimulate ECAR and glucose uptake by BMDMs derived from *Gpr3*^{-/-} mice (Fig. 2d-e) or ImKCs deficient in GPR3 through CRISPR-cas9 knockout or siRNA inhibition (Supplementary Fig. 2e-j). The reviewer raised an important question as DPI might have multiple targets as 500 nM DPI was able to stimulate ECAR by *Gpr3*^{-/-} BMDMs, as well as *Arrb2*^{-/-} or *Pkm2*^{-/-} ImKCs (Fig. 2d, 2g-h, 3b-c). Clearly, DPI target specificity requires further investigation.

References:

Capaldi et al. 2018, Scientific Reports 8:11102-
Mahmud et al. 2017, Scientific Reports 7:17953-

2. In one strand of the work the authors employ cells from p47phox^{-/-} mice to try to eliminate a potential non-GPR3-mediated effect of DPI. However, surprisingly, they do not use GPR3^{-/-} mice although such animals were first described a number of years ago (e.g. Sci Rep 2015 Oct 12;5:14953 and Sci Transl Med 2015 Oct 14;7(309):309ra164). The authors need to explain why they did not use such animals.

Response: Per reviewer's suggestions, we have tried to obtain *Gpr3*^{-/-} mice. However, due to pandemic, we were only able to obtain bone marrow cells from *Gpr3*^{-/-} mice from Dr. Grzegorz Godlewski (NIAAA/NIH). *Gpr3*^{-/-} bone marrow-derived macrophages failed to respond to 50 nM DPI by upregulating ECAR, consistent with results from ImKCs deficient in GPR3 through CRISPR-cas9 knockout or siRNA inhibition (Fig. 2d-e and Supplementary Fig. 2e-j). We have added the new results from *Gpr3*^{-/-} BMDMs and *Gpr3*^{-/-} ImKCs in the revised **Fig. 2d-e**, and **Supplementary Fig. 2**.

3. Even if there is a valid reason to not use GPR3^{-/-} mice then the studies with DPI need to be augmented with the concurrent use of a GPR3 inverse agonist. In many ways these are as limited as the agonist ligands but molecules such as AF64394 (Bioorg Med Chem Lett. 2014 Nov 15;24(22):5195-8, and J Biomol Struct Dyn. 2021 Jun 23:1-10. doi: 10.1080/07391102.2021) would seem to offer an opportunity to provide greater confidence in the identification of GPR3 as the relevant molecular target.

Response: Since we addressed the above questions with BMDMs derived from *Gpr3*^{-/-} mice and *Gpr3*^{-/-} ImKCs using CRISPR-Cas9-mediated gene editing we did not use the suggested inverse agonists as they are not ideal either.

4 The studies on the immortalised Kupffer cells (ImKCs) are interesting. However, once again in terms of defining the role of GPR3 why is GPR3 'knocked-down' in these cells whilst beta-arrestin 2 is 'knocked-out'? If the gene-editing protocols work effectively in these cells surely it would be most effective to eliminate GPR3 expression?

Response: We thank for the reviewer's suggestions. We have now generated GPR3 knockout ImKCs by CRISPR-cas9 gene editing and tested the effects of DPI. Consistent with results from *Gpr3*^{-/-} BMDMs and GPR3 knockdown (siRNA) ImKCs, DPI (50 nM) did not stimulate ECAR

and glucose consumption in GPR3-null ImKCs. The new results are included in the **Supplementary Fig. 2** of the revised manuscript.

5. In a small extension to this point why specifically eliminate beta-arrestin 2? Is beta-arrestin 1 also expressed by the cells? These isoforms certainly have overlapping functions and in other cell systems (e.g. HEK293 cells) researchers have selected to knock-out both isoforms and then potentially reconstitute function by selective re-introduction of either isoform.

Response: As shown in **Figure X1** below, β -arrestin 2 is expressed at much higher level than β -arrestin 1 in both mouse and human macrophages based on single-cell RNAseq analysis. Although GPR3 was co-IPed with both arrestins in 293T cells where both GPR3 and arrestins were overexpressed (Ref. 21), it has been shown that GPR3 preferentially interacts with β -arrestin 2 during its action (Ref. 26, Ref. 21 and 22). To further address the reviewer's concern, we performed the colocalization assay of GPR3 and β -arrestin 1 in ImKCs. DPI did not induce the colocalization of GPR3 and β -arrestin 1 (Fig. X1C). Furthermore, translocation assay showed DPI did not induce the translocation of β -arrestin 1 to the cell membrane in ImKCs while DPI induced β -arrestin 2 translocation to the cell membrane (Fig. 2i and Supplementary Fig. 2n-o). Importantly, DPI's effect (at 50 nM) on glycolysis was completely abolished in β -arrestin 2 knockout cells, suggesting that β -arrestin 1 does not play any significant role.

Figure X1. A, ARR1 and ARR2 expression in human immune cells from Single Cell Portal from <https://singlecell.broadinstitute.org>. B, Arr1 and Arr2 expression in mouse immune cells from Mouse cell atlas (<http://bis.zju.edu.cn/MCA/search.html>). C. Localization of GPR3 and β -arrestin 1 in ImKCs treated with DMSO or DPI (50nM) for 6 hrs. GPR3-myc and β -arrestin1-GFP (ARR1) were transfected into ImKCs and then treated with DPI or DMSO. Cells were fixed and stained with anti-Myc and anti-Arr1.

Minor Point

In supplementary Figure 2 panel c it would be important to show the full immunoblot corresponding to GPR3. Most GPCRs do not migrate as a tight single band in SDS-PAGE due to differential N-glycosylation. The appearance of such a single band is thus rather concerning in terms of potential specificity of the antiserum. No molecular mass markers are provided for these immunoblots and are lacking in a number of other cases.

Response: We thank the reviewer's comment. We noticed that when we increased the amount of lysate loaded per lane and length of time of electrophoresis, we observed a faint band above the main GPR3 band. This data is included in the revised **Supplementary Fig. 2e** with parental (WT), CRISPR-cas9 knockout and siRNA knockdown ImKCs for comparison.

Reviewer #2 (Remarks to the Author):

In this manuscript entitled 'Activation of GPR3-β-Arrestin2-PKM2 pathway in Kupffer cells protects against high-fat diet induced obesity and liver pathogenesis through enhanced glycolysis' the authors study metabolic regulation of hepatic macrophages during HFD-induced obesity. By activating the glycolytic rate in macrophages, the authors demonstrate a protection against the harmful metabolic effects of obesity and NAFLD. The authors have used an extensive approach by including many different experimental setups ranging from in vitro to in vivo animal and human studies.

I feel that the current manuscript may lack some balance with a lot of focus on the mechanistic pathways leading to activation of glycolysis by DPI. Although these results are very important, most likely other pathways would be involved in explaining the in vivo observations with a reduction in BW and improvement in glucose control upon DPI treatment. Although the absence of PKM2 seems to abolish most of the effects of DPI treatment in vivo, a potential underlying mechanisms to explain the metabolic effects is currently lacking. This is even more important since an increase in glycolytic rate and a strong reduction in oxidative phosphorylation after DPI treatment resembles the Warburg effect and is linked to pro-inflammatory activation of macrophages. However, the results in the manuscript show opposite effects of DPI treatment on inflammation. This is hugely interesting, however, it requires careful interpretation and needs to be discussed in light of the current literature that points to glycolysis as a driver of inflammation.

I do have several other comments and suggestion that may help to further improve the quantify of the current manuscript.

Response: We thank the reviewer for the constructive comments and have revised the manuscript accordingly to address the following comments. As the reviewer correctly pointed out, our results show that DPI stimulates glycolysis and glucose consumption in mice without inducing inflammatory cytokine expression. This is different from general understanding that inflammatory macrophages glycolysis for energy metabolism. Our findings that DPI stimulates glycolysis without stimulating inflammatory cytokine production suggests that the two processes (glycolysis and cytokine production) are not mechanistically linked but separable. This interpretation is consistent with our previous findings that DPI does not upregulate the gene modules which are

upregulated by IFN γ or LPS (Ref. 15), but suppresses the reactive oxygen species (ROS) pathway, which is upregulated by LPS. Similarly, the pentose phosphate pathway (PPP) and ROS are required for inflammatory cytokine production (Ref 44 and 45), however, DPI does not upregulate the expression of G6PD that is a key enzyme for switching to PPP (Ref. 15). We have added these discussions in the revised manuscript.

1. *Figure 1A has not much added value. The metabolic effects of DPI treatment, especially at the highest concentration, are very strong. It seems to almost completely stop oxygen consumption. Any data available on cell viability?*

Response: Per reviewer's suggestion, we moved **Fig.1a** to supplementary data as **Supplementary Fig. 1a**. To investigate cellular cytotoxicity of DPI, ImKCs were seeded into 96-well plates in triplicates and cultured for 12 h and then were treated with DMSO or DPI (50 nM and 500 nM) for 24 hrs. Cell viabilities were assayed by CCK8 and MTT methods (Beyotime, China). As shown in **Figure X2**, DPI at either 50 nM or 500 nM DPI for 24hrs did not induce significant cell death of ImKCs.

Figure X2. The cellular cytotoxicity of DPI in vitro.

2. *Have the authors looked at the effects of LPS stimulation and DPI treatment? LPS is known to rely on an increase in glycolysis to induce a pro-inflammatory response. In addition to studying the mechanism of action in figure 1-4, I would be very interested in functional readouts of the cells. How does this treatment impact on the functional properties of the macrophage (migration, ROS production, cytokine production, phagocytosis) ? This would be crucial to understand the in vivo effects as well.*

Response: We presented these data in our previous publication (Ref. 15), in which it was shown that DPI (500 nM) polarizes hMDM to M1-state. However, as shown in **Figure X3A** from the supplementary data in Ref.15, unlike other M1-like compounds, DPI does not upregulate gene modules which are upregulated by IFN γ (#7&8) or LPS (#30-32), but suppresses the gene modules which also are downregulated by LPS (#26&27). It seems that DPI at 500 nM (used in Ref. 15) only partially induced M1-biased polarization in hMDMs *in vitro*. Further validations have shown that DPI does not promote the expression of M1 markers/cytokines but suppresses M2 markers/cytokines at the transcriptional and protein levels (**Figure X3B**). Although in general proinflammatory macrophages use glycolysis for energy metabolism, the switch to pentose phosphate pathway (PPP) is necessary for their cytokine production (see reviews of Ref. 44 and 45), however, DPI does not upregulate the expression of G6PD that is a key enzyme for switching to PPP. Our findings that DPI stimulates glycolysis without stimulating inflammatory cytokine

production suggests that the two processes (glycolysis and cytokine production) are not mechanistically linked but separable. We have revised the Discussion to include these points.

Figure X3. The effect of DPI on macrophage phenotype. A. GSEA of macrophage activation modules from Fig. 4c (Hu et al. 2021, Ref.15). B. Gene expression and protein expression changes of polarization markers and cytokines induced by DPI in hMDMs quantified by RNAseq, qPCR and flow cytometry (Supplementary Fig. 5 in Ref 15). Red arrow points to DPI treatment.

Reference:

Ref. 15: Hu et al. 2021, Nature communications, 12:773

Ref. 44: Liu et al. 2021, Biomark Res: 9:1. <https://doi.org/10.1186/s40364-020-00251-y>

Ref. 45: Viola et al. 2019, Frontiers in Immunology, 10:1462

3. The animal studies are interesting, yet lack several crucial readouts. The authors should quantify TG content in the liver. Other metabolic parameters should be measured including insulin. I am surprised by the data in figure 5c. The HFD seems not to increase eWAT while the increase in total BW is very robust after HFD-feeding. Also, labelling of the different groups is incorrect in figure 5c.

Response: Per reviewer's suggestion, we have now quantified the blood/plasma insulin levels and triacylglycerol (TG) contents in the liver 8 weeks post HFD with vehicle or DPI treatment or in mice on normal chow. Results are consistent with the changes of body weight and fat mass shown in **Fig. 5**: i.e., DPI-treated HFD mice had decreased levels of blood insulin and liver TG compared to vehicle-treated HFD mice. We included this data in the **Supplementary Fig. 6a-b**. In **Fig. 5c**, the original labels had mistakes, including the wrong legend and switch between eWAT and iWAT. We have corrected these errors in **Fig. 5c** of the revised manuscript.

For the *in vivo* treatment with DPI, I would also be very interested in effects in other tissues. By using IP injections, the compound most likely would affect many other cells including macrophages residing in different tissues. Keeping in mind the very robust effects on glycolysis presented in the first part of the manuscript, other cells/tissue most likely would be affected by the

treatment. It would be important to include effects of the treatment in other cells/tissues as well. The in vivo effects can not be explained by an effect on liver macrophages alone. Although effects on BW/liver TGs are abolished in the knockout model (figure 5g/h), other metabolic parameters are not presented in the manuscript. These data are needed to be able to fully interpret effects of the in vivo treatment with DPI.

It seems the authors are unable to present a mechanism to explain the in vivo observations. Also, a critical reflection using existing studies on the effects of glycolysis on macrophage activation, is lacking.

Response: We thank for the reviewer's constructive comments. We performed several additional experiments to address the reviewer's concerns. Firstly, we examined the DPI effects on other cell types including primary hepatocytes and 3T3-L1 differentiated adipocytes. As shown in the **Supplementary Fig. 8a-b** of the revised manuscript, like in *Gpr3^{-/-}*, *Arb2^{-/-}* or *Pkm2^{-/-}* macrophages, DPI at 50 nM did not have any effects on ECAR while at 500 nM slightly increased the ECAR in both hepatocytes and adipocytes. Secondly, as shown in the **Supplementary Fig. 6c** of the revised manuscript, H&E staining of eWAT tissues showed that the sizes of adipocytes in eWAT were smaller from DPI-treated HFD mice (8 wks) than vehicle-treated HFD mice. Importantly, we did not see significant eWAT browning from DPI-treated HFD mice, suggesting DPI inhibits the lipid accumulation in eWAT without activating WAT.

GPR3 is lowly expressed in WAT even under the cold condition (Ref. 19, Johansen et al. 2021). Interestingly, we observed that the HFD-fed KC-specific *Pkm2^{-/-}* mice treated with DPI had less eWAT deposit (**Supplementary Fig. 9b**), suggesting DPI might affect WAT directly or indirectly through other cells. Although via Gs-coupling and cAMP activation, cold-induced GPR3 activation could suppress BW gain by increasing thermogenesis of BAT not WAT, transcriptional analysis of BAT GPR3 knock-out or knock-in demonstrated that GPR3 signaling in BAT is independent of PKM2 pathway.

Taken together, DPI induced GPR3-PKM2 activation in KCs is sufficient to prevent the HFD-induced liver pathology and may not directly involve other types of cells in vivo based on the following observations: 1) PKM2 is highly expressed in liver macrophages but not hepatocytes and adipocytes, 2) DPI at 50 nM does not increase ECAR of hepatocytes or adipocytes; 3) liver pathology is not prevented by DPI in the KC-specific *Pkm2^{-/-}* mice fed with HFD but still prevented eWAT deposit; and 4) PKM2-null mice are more sensitive to HFD induced liver pathology.

We have revised the Discussion to address the concerns raised by the reviewer.

Reviewer #3 (Remarks to the Author):

Dong et al. studied the effects of DPI on metabolic reprogramming in both mice models and human patients. They showed the activation of GPR4-β-Arrestin2-PKM2 pathway in Kupffer cells could prevent HFD-induced obesity and liver pathogenesis. In addition, they identified a KC population related to disease (DAM) by using single-cell RNA-seq and demonstrated the reversion of glycolysis and inflammation associated aberrations after DPI treatment in vitro. However, the bioinformatics data analysis needs further concerns.

Response: We thank the reviewer for the constructive comments and have revised the manuscript accordingly to address the following comments.

Major

1. In L336, the authors said “C3 was the only elevated LM population” which may refer to Fig. 7b. However, the elevation is largely contributed by NAFLD1, and there is no evident increase of C3 cells in other two patients. In addition, the display of cellular composition alteration in Fig. 7b is not appropriate, which is highly sensitive to the number of cells harvested in each patient. Instead, it is better to calculate the ratio of observed to expected number of cells in each patient and test the significance of C3 increase as compared to healthy donors.

Response: We agree with the reviewer’s constructive comments. We have removed “C3 was the only elevated LM population” from revised manuscript to avoid the over interpretation. Per reviewer’s suggestion, we calculated the ratio of observed to expected number of cells for each cluster in each patient using Ro/e chi-square test (Guo et al. 2018). There is no significant difference in different macrophage subclusters between NAFLD patients and healthy donors. However, the macrophage population in total liver immune cells is significantly increased in NAFLD patients as compared to healthy controls, consistent with previous publications (see **review of Ref. 4**). This data is included in the **Supplementary Fig. 10 and 11** of the revised manuscript.

Reference

Guo et al. 2018, Nature medicine, 24:978

2. Are there any transcriptional alterations in KCs other than C3? In addition to single-cell RNA-seq, this work has several bulk data sets which support most of the clues about gene expression changes related to glycolysis and inflammation. However, C3 cells are a small portion of KCs in patients according to scRNA-seq. Therefore, it is necessary to compare the transcriptomes of other KCs in patients to healthy donors.

Response: Per reviewer’s suggestions, we performed additional analysis of the transcriptional changes in macrophage subclusters between NAFLD and healthy donors. Overall, macrophages in NAFLD upregulated pathways of antigen presentation and inflammatory response (Supplementary Fig. 11). Since C5 (exhausted KC) and C6 (RBC-phagocytosing) consist of very small proportions, we focused to identify the upregulated DEGs in C0 (KC), C1 (moMac), C2 (moMac), C3 (DAM) and C4 (DC-like) between NAFLD and healthy donors. GO enrichment analysis of the DEGs showed consistently that different clusters upregulated pathways of inflammatory response or NFκB activity or response to LPS, although each cluster had upregulated specific pathways. The most dramatical alterations of pathways are in C3 and C4. We have incorporated these results into the revised manuscript.

3. Though this study is mainly about macrophages, it is also interesting to investigate the whole immune environment by in-depth analysis of other immune cells in patients as the authors performed single-cell RNA-seq.

Response: Per reviewer's suggestions, we analyzed transcriptional changes in different types of immune cells between NAFLD and healthy donors. As shown in the **Supplementary Fig. 10** of the revised manuscript, GSEA analysis showed although different cell types up- or down-regulated various pathways, metabolic pathways including mTOR, glycolysis, cholesterol and fatty acid were consistently altered in different cell types. Since the focus of this study is to on DPI's effects on macrophages, we included this data in the **Supplementary Fig. 10** and briefly described it in the result section.

Minor

1. L918, "expression profile in a with the", a typo error or missing words after "in a".

Response: We corrected this error with "6a" as "a" refers to Fig. 6a.

2. Fig. 7d is not informative as it is presented this way. A combination of Fig. 7c and d should be better. Otherwise, Fig. 7d should be removed or moved to the sup figure to just indicate a unique expression in each cell cluster.

Response: We agree with reviewer's suggestion and moved **Fig. 7d** to **supplementary Fig. 11**.

3. Fig. 7e, cells in C5 and C6 are not shown, is it intended or mistake? It is not clear about the meanings of colors, do colors indicate pseudotime? Details should be added to the legend.

Response: C5 and C6 clusters are present in the figure (dots in the bottom in old Fig 7e), but invisible because of resolution and small number of cells. The pseudotime color is amended into tSNE plot as Fig. 7a. We corrected this figure (now the **Fig.7c**) with higher resolution and color keys of pseudotime in the revised manuscript.

4. Sup. Fig. 9, figure legend, the description "relative proportion of each cluster in each sample (f)" is not correct as (f) displays the composition of donor-derived cells in each cluster instead of in each sample. Again, this type of display is not appropriate.

Response: Per reviewer's suggestion, we calculated the ratio of observed to expected number of cells in each patient use Ro/e chi-square test to replace the relative proportion in the revised manuscript.

5. More details about the single-cell RNA-seq data processing are needed in the method section. For instances, batch correction, the rationale of 15 PCs selection, the threshold of determining differentially expressed genes, procedures of performing trajectory inference.

Response: Per reviewer's suggestion, we have added more detailed description of data processing, PC selection determined by the sampling test with JackStraw procedure, threshold for DEG and trajectory inference in the method in the revised manuscript.

6. The gene lists used to perform GSEA should be summarized to a sup table providing the details of references where the genes are collected.

Response: GSEA analysis is based on entire list of gene detected by RNAseq. We updated the method section with more details and provided the Supplementary Tables for the DEGs and the gene list for GSEA in the revised manuscript.

Reviewer #4 (Remarks to the Author):

The manuscript of Dong et al is an extensive analysis on the role of DPI on the metabolic reprogramming of macrophages and its underlying molecular mechanism. The work is quite comprehensive, however, there are deficiencies in detecting metabolic flux and PKM2 status. The manuscript is not suitable for publication until these issues are resolved.

Response: We thank the reviewer for the constructive comments and have revised the manuscript accordingly to address the following comments.

Introduction

"GPR3 is highly expressed in the brain...neurological processes" The authors did not study the function of GPR3 in the nervous system. The authors should focus here on the research progress of DPI, GPR3 or β -arrestin2 in NFLAD and Kupffer cells, especially on metabolic regulation.

Response: We thank the reviewer's suggestions, and we updated the abstract section by emphasizing the role of GPR3 in the metabolism in the revised manuscript.

Results

Figure 1: The authors confirmed that DPI promotes the conversion of TCA cycle to glycolysis in glucose metabolism, which is manifested as an increase in ECAR value, an increase in glycolytic metabolites, and a decrease in TCA metabolites. They should first consider whether DPI affects glucose uptake from the extracellular environment. They should identify the difference in metabolite content between DPI-treated cells and control cells under the same amount of glucose uptake (Fig.1f).

Response: To address the reviewer's concerns, we measured the ECAR values of ImKCs treated for 10 min with DPI in the presence or absence of glucose. As shown in **Supplementary Fig. 1d** of the revised manuscript, DPI similarly stimulated ECAR (glycolysis) with or without extracellular glucose, suggesting that DPI-induced immediately increase in glycolysis relies on activation of glycolytic enzymes rather than glucose uptake.

Of course, glucose uptake will increase when the intracellular glucoses are consumed. We don't think it is reasonable to remove extracellular glucose to assay the metabolites in DPI-treated cells. Under such a condition, when intracellular glucose is consumed, and metabolites would decrease, and cells would adopt to other metabolism pathways.

Figure 2: Fig.2c is not mentioned in the text.

Response: We cited the wrong figures for Fig. 2b and 2c. We have corrected these in the revised manuscript.

Figure 3: The authors confirmed that DPI promotes the interaction of PKM2 and β -arrestin2, thereby enhancing glycolysis. Does β -arrestin2 interact with PKM1? We know PKM1 and PKM2 play opposite roles in glucose metabolism. The PKM1 and PKM2 proteins are derived from the same precursor PKM mRNA and differ only by 22 amino acids. Although PKM2 is the main pyruvate kinase subtype in immune cells, tumor cells or embryonic stem cells, the precursor PKM mRNA tends to form PKM1 under some stimulation. They should consider whether DPI affects the alternative splicing of PKM mRNA.

Response: To address the reviewer's concern, we performed co-IP of β -arrestin2 with PKM1 or PKM2 in ImKCs. Since endogenous PKM1 could not be detected by immunoprecipitation of β -arrestin2 in ImKCs due to the low expression, we performed the immunoprecipitation by co-transfecting β -arrestin2 plus either PKM1 or PKM2 with HA-tag in ImKCs and 48 hours later treated the cells with or without 50 nM DPI for 6 hrs. Cell lysates were immunoprecipitated with anti- β -arrestin2 and the precipitates were analyzed by Western blotting. As shown in **Supplementary Fig. 7d** in the revised manuscript, β -arrestin2 coimmunoprecipitated with both PKM1 and PKM2. However, DPI only induced the interaction of β -arrestin2 with PKM2. Consistently, DPI did not stimulate the expression of *Pkm1* isoform but stimulated the expression of *Pkm2* isoform in ImKCs at the transcriptional level (**Supplementary Fig. 7c**). Furthermore, knockout of the PKM2 isoform specifically in BMDMs abolished the effect of DPI. Taken together, DPI-induced glycolysis is dependent on PKM2 but not PKM1, consistent with the predominant expression of PKM2 in Kupffer cells.

Figure 4: Under DPI treatment, the monomer, dimer and tetramer of PKM2 all appeared to increase, compared with DMSO controlled cells (Fig.4b). Does this mean that DPI directly promotes the expression of the entire PKM2 protein? Due to the high expression of PKM2 caused by DPI treatment, the authors observed an increase in PKM2 bound to β -arrestin2 protein and an increase in the formation of PKM2 dimers. Therefore, the western blotting data cannot support the authors' conclusion that DPI induces the formation of the dimer PKM2.

Response: To address the reviewer's concern, we quantified the PKM2 isoforms in ImKCs treated with DPI for 1 hr by Western blotting. As shown in **Fig. 4b-c** of the revised manuscript, PKM2 dimer increased while PKM2 monomer decreased following 1 hr DPI treatment. Since 1 hr treatment is unlikely to significantly increase the total PKM2 protein level by stimulating mRNA transcription and protein translation, this data would suggest that DPI directly stimulates the formation of PKM2 dimer.

Figure 4: The authors confirmed the translocation of PKM2 from the cytoplasm to the nucleus by immunofluorescence (Fig.4c). It is also necessary to isolate the nuclear and cytoplasmic components of the KC cells, and detect the expression of nuclear PKM2 and cytoplasmic PKM2 by western blotting.

Response: Per reviewer's suggestions, we isolated proteins from the nuclear and cytoplasmic fractions of ImKCs treated with DMSO or 50 nM DPI to detect the PKM2 expression by Western blotting. As shown **Fig. 4d** of the revised manuscript, DPI only induced the increase of nuclear PKM2 in ImKCs.

REVIEWER COMMENTS

Reviewer #1 (Remarks to the Author):

Despite the inability to perform some of the additional control experiment suggested, not least the authors have failed to access and use GPR3 knockout mice, the authors have made a substantial effort, including some genome editing studies, to perform additional studies that give comfort to the hypothesis that the effects reported are transduced, at least in larger part, via GPR3. Whilst less than ideal, overall this is a substantial and interesting piece of work.

Reviewer #2 (Remarks to the Author):

The manuscript has really improved. However, I still have several concerns. No data is provided related to the inflammatory output of macrophages after treatment with DPI. This is only presented in the RNAseq experiments. It would be important to further functionally characterize the macrophages after DPI treatment, especially since the authors show that enhanced glycolysis would not lead to more inflammation. Also, the authors suggest a key role of macrophages in mediating various metabolic effects of DPI in vivo making it even more important to further characterize effects of DPI on macrophages beyond metabolism. Also, I feel that the in vivo experiments require some more work, especially in describing and presenting the results. More relevant data related to metabolic parameters is now shown for the WT model, however, not all of these parameters are shown for the KO model. It would be nice to end the discussion with a more general paragraph and overall conclusion.

Reviewer #3 (Remarks to the Author):

The authors addressed all the concerns. I have no more comments.

Reviewer #4 (Remarks to the Author):

The authors answered the question comprehensively, and I have no more comments.

REVIEWER COMMENTS

Reviewer #2 (Remarks to the Author):

The manuscript has really improved. However, I still have several concerns. No data is provided related to the inflammatory output of macrophages after treatment with DPI. This is only presented in the RNAseq experiments. It would be important to further functionally characterize the macrophages after DPI treatment, especially since the authors show that enhanced glycolysis would not lead to more inflammation.

Response: In response to reviewer's concerns, we have performed additional studies to demonstrate that DPI does not stimulate macrophage inflammatory responses. First, we assayed the effects of DPI on phagocytosis, ROS and cytokine/chemokine production by human monocyte-derived macrophages (hMDMs). For these experiments, monocytes were purified from PBMCs from four healthy donors and cultured in the presence of M-CSF to generate hMDMs. To assay the effect of DPI on macrophage phagocytosis, hMDMs were treated with 50 or 500 nM DPI for 24 hrs and then incubated with 15 μ g/mL pHrodo Green *E. coli* bioparticles for 1 hr. Phagocytosis was quantified by flow cytometry. As shown in **Supplementary Fig. 10d**, DPI stimulates phagocytosis of *E. coli* bioparticles by hMDMs.

To assay the effect of DPI on macrophage ROS production, hMDMs were treated with 50 or 500 nM DPI for 24 hrs and then incubated with 5 μ M CM-H2DCFDA and total ROS were quantified by flow cytometry. As shown in **Supplementary Fig. 10c**, DPI inhibits production of intracellular ROS in hMDMs.

To assay the effect of DPI on macrophage production of cytokines and chemokines at the protein level, hMDMs were treated with DMSO or 50 nM DPI for 24 hrs. Equal amount of culture supernatant from four PBMC donors was mixed and applied to the human cytokine antibody array specific for 60 targets (Abcam, #Ab169817). As shown in **Supplementary Fig. 10a**, DPI does not significantly stimulate hMDMs to produce inflammatory cytokines.

Furthermore, we assayed the effect of DPI on cytokine and chemokine production in mice. For this experiment, B6 mice (n=4) were dosed with 2 mg/kg DPI i.p. and plasma was collected before dosing and 48 hrs after dosing. Equal amount of plasma from four mice at the same time-point was mixed and applied to the mouse cytokine antibody array specific for 62 targets (Abcam, #Ab133995). As shown in **Supplementary Fig. 10b**, compared to the plasma collected before DPI treatment, only two anti-inflammatory cytokines CCL9 and CXCL4 were visibly elevated following DPI treatment; none of the proinflammatory cytokines were elevated. Therefore, DPI does not stimulate production of inflammatory cytokines in mice.

These new results are consistent with our RNAseq results, which show a lack of transcripts encoding inflammatory cytokines in macrophages after DPI treatment, as well as the results that DPI does not increase the expression of M1 markers (CD86 and CD80) but suppresses the expression of CD206 (see the first point-to-point response and our previous publication [reference 15]).

The new results are included in the revised manuscript as Supplementary Fig. 10. The relevant texts have been added in the Results, Discussion, and Material and Method sections.

Also, the authors suggest a key role of macrophages in mediating various metabolic effects of DPI in vivo making it even more important to further characterize effects of DPI on macrophages beyond metabolism. Also, I feel that the in vivo experiments require some more work, especially in describing and presenting the results.

Response: See responses above. We have made some edits of the text to enhance the clarity of description and presentation of the in vivo experiments and results.

More relevant data related to metabolic parameters is now shown for the WT model, however, not all of these parameters are shown for the KO model. It would be nice to end the discussion with a more general paragraph and overall conclusion.

Response: To address the reviewer's concern, we have performed metabolic profiling of KO cells. WT and *Gpr3^{-/-}* ImKCs were treated with DPI (50 nM) for 24 hrs and the select metabolites were quantified by LC-MS. As shown in **Supplementary Fig. 1f**, DPI at 50 nM failed to stimulate any significant increase in glucose consumption and glycolytic intermediates in *Gpr3^{-/-}* ImKCs, while consistently stimulated the decrease of intracellular glucose, TCA cycle intermediates and the increase of glycolytic intermediates in wildtype ImKCs. Similarly, *Pkm2^{-/-}* BMDMs were treated with DMSO or DPI (50 nM and 500 nM) for 24 hrs and the select metabolites were quantified by LC-MS. As shown in **Supplementary Fig. 1g**, DPI did not stimulate any significant increase in glucose consumption and glycolytic intermediates in *Pkm2^{-/-}* BMDMs. These results are included in the revised manuscript as **Supplementary Fig. 1f-g**.

We have also revised the Discussion with a more general paragraph and overall conclusion at the end.

REVIEWER COMMENTS

Reviewer #5 (Remarks to the Author):

The manuscript by Dong et al provides intriguing insights into the possible contribution of diphenylethylideneiodonium (DPI), an agonist of G-protein coupled receptor 3 (GPR3), in shaping cellular metabolism and mainly glycolysis, and how such outcomes regulate high-fat diet (HFD)-induced obesity and liver pathogenesis. As an ad hoc reviewer for this manuscript, the overall assumption is that concerns raised initially with focus on macrophage inflammatory and metabolic function remain. Specific points are provided below.

1. The concerns regarding macrophage-specific inflammatory output remain. Despite the authors attempt to enumerate phagocytosis, ROS production and inflammatory mediators from human MDM and in serum of in vivo treated mice, the quantification of macrophage specific production of dogmatic mediators of inflammation (e.g., TNF, IL-6, IL-1 etc.) are missing. Addition of such results would strengthen the findings and the conclusions stated. Furthermore, there is a concern with Supplemental Fig 10a and b where both human and mouse samples are mixed (n=4) into a single well of inflammatory markers. Notably, such approach limits the detection of individual differences and completion of adequate statistical analyses.
2. The concerns regarding additional analyses in vivo remain. Additional studies focused on functional analyses of liver isolated macrophage or KC of chow or HFD fed animals with or without DPI have not been completed. Lack of such analyses restricts our findings about the effect of DPI on macrophage/KC function and is merely based on the RNAseq analysis.
3. The authors have partly addressed concerns regarding additional analyses of metabolic parameters of their genetic-deficiency models. New data from Gp3 KO ImKCs versus WT is included and as such now strengthens the study conclusions. However, the authors did not provide metabolic analyses in PKM2-deficient KCs cells, instead here they only focused on PKM2-deficient BMDMs. As tissue function and inflammatory milieu has a strong impact on cellular phenotype/function, assumption that BMDMs behave in the same way as KCs is not sufficient. Furthermore, the comparison of PKM2-deficient BMDMs were to WT BMDMs

should be shown. Lastly, with PKM2 being downstream of glycolytic steps examined in BMDMs (e.g., G6P) and total glucose analyses additional studies and discussions are needed to fully demonstrate how PKM2 shapes KC function. we questioned whether cells have functional metabolic pathways.

Reviewer's comments are *in italics* and the responses are in regular blue font.

Reviewer #5 (Remarks to the Author):

1. The concerns regarding macrophage-specific inflammatory output remain. Despite the authors attempt to enumerate phagocytosis, ROS production and inflammatory mediators from human MDM and in serum of in vivo treated mice, the quantification of macrophage specific production of dogmatic mediators of inflammation (e.g., TNF, IL-6, IL-1 etc.) are missing. Addition of such results would strengthen the findings and the conclusions stated.

Response: Per reviewer's suggestions, we have purified liver macrophages using anti-F4/80 beads from B6 mice, treated them with DMSO or DPI at 50nM or 500nM for 24hrs, and then stimulated them with 100ng/mL LPS for 6 hours in fresh medium without DPI. The production of ROS was measured by flow cytometry and the production of proinflammatory cytokines TNF- α , IL-1 β and IL-6 in the supernatants were quantified by ELISA. Results showed that DPI pre-treatment suppressed ROS production and secretion of TNF- α , IL-1 β and IL-6 in KCs under LPS stimulation. These data are included in the revised manuscript as Supplementary Fig. 11a-b.

Furthermore, there is a concern with Supplemental Fig 10a and b where both human and mouse samples are mixed (n=4) into a single well of inflammatory markers. Notably, such approach limits the detection of individual differences and completion of adequate statistical analyses.

Response: In the original "Supplemental Fig 10a and b" (now Supplementary Fig.11c-d of the revised manuscript), human samples and mouse samples were assayed separately. We mixed equal amounts of culture supernatants from four hMDM cultures for array analysis. Separately, we mixed equal amounts of plasma from four different mice for array analysis. Human MDM culture supernatants and mouse plasma were NOT mixed together for array analysis. We agree with the reviewer that the mixing limits the detection of individual differences and completion of adequate statistical analyses. Our analysis was intended to broadly survey for cytokines (>60) that may be induced by DPI in mice and in human MDM. Considering that we have now analyzed inflammatory cytokine and ROS production in KCs isolated from normal B6 mice (see below), HFD-fed mice, with or without DPI treatment, we hope that the reviewer would agree with us that the results from the array analysis becomes less important. We have revised the legends and text to make it clearer.

2. The concerns regarding additional analyses in vivo remain. Additional studies focused on functional analyses of liver isolated macrophage or KC of chow or HFD fed animals with or without DPI have not been completed. Lack of such analyses restricts our findings about the effect of DPI on macrophage/KC function and is merely based on the RNAseq analysis.

Response: Per reviewer's suggestion, we have now directly assayed the effect of DPI on the production of inflammatory cytokines and reactive oxygen species (ROS) in purified KCs from vehicle- or DPI-treated HFD-fed mice and age-matched mice fed with NC. Purified KCs were not stimulated or stimulated with LPS for 6 hours and cytokines in the culture supernatants were assayed by ELISA and intracellular ROS were detected by flow cytometry. LPS stimulated

production of pro-inflammatory cytokines TNF- α , IL-1 β and IL-6 in KCs, but the levels were significantly lower in KCs from DPI-treated mice than those of vehicle-treated mice, reaching the similar levels as in mice fed with NC. LPS-induced ROS production was also significantly reduced in KCs purified from HFD-fed DPI-treated mice as compared to HFD-fed vehicle-treated mice. These new data are consistent with the RNAseq results and are shown as **Fig. 6e-f** of the revised manuscript.

Additionally, we have assayed ROS production and phagocytosis of hMDMs following DPI treatment, and found that DPI suppressed ROS production but stimulated phagocytosis. These data are included in the revised manuscript as Supplementary Fig. 11e-f.

3. The authors have partly addressed concerns regarding additional analyses of metabolic parameters of their genetic-deficiency models. New data from Gp3 KO ImKCs versus WT is included and as such now strengthens the study conclusions. However, the authors did not provide metabolic analyses in PKM2-deficient KCs cells, instead here they only focused on PKM2-deficient BMDMs. As tissue function and inflammatory milieu has a strong impact on cellular phenotype/function, assumption that BMDMs behave in the same way as KCs is not sufficient.

Response: Per reviewer's suggestion, we isolated KCs from livers of *Pkm2*^{-/-} mice and *Pkm2*^{fl/fl} control mice as well as age-matched normal B6 mice to measure metabolic response to DPI by the seahorse analysis. DPI stimulated a significant increase in ECAR, glycolytic capacity and glycolytic reserve of KCs isolated from B6 mice and *Pkm2*^{fl/fl} mice in a dose-dependent manner. However, DPI at 50 nM failed to stimulate any significant increase in ECAR, glycolytic capacity and glycolytic reserve of *Pkm2*^{-/-} KCs, similar to *Gpr3*^{-/-} or *Arrb2*^{-/-} ImKCs and *Gpr3*^{-/-} BMDMs. Furthermore, we purified KCs from vehicle- or DPI-treated HFD-fed *Pkm2*^{fl/fl} and *Pkm2*^{-/-} mice and directly measured their basal level of ECAR. ECAR was significantly higher in KCs from *Pkm2*^{fl/fl} mice that were treated with DPI than vehicle. In contrast, ECAR was significantly lower in KCs from *Pkm2*^{-/-} mice and DPI treatment did not increase ECAR. These results show that DPI likely stimulates glycolysis in KCs in mice through the same mechanisms. These new data are shown as **Fig. 5i** and Supplementary Fig. 10 of the revised manuscript.

Furthermore, the comparison of PKM2-deficient BMDMs were to WT BMDMs should be shown.

Response: These data are shown in the original manuscript as Supplementary Fig. 1e-f. In the revised manuscript, they are shown as Fig. 3b-c, Supplementary Fig. 4b-c, f.

Lastly, with PKM2 being downstream of glycolytic steps examined in BMDMs (e.g., G6P) and total glucose analyses additional studies and discussions are needed to fully demonstrate how PKM2 shapes KC function. we questioned whether cells have functional metabolic pathways.

Response: In our study, we show that *Pkm2*^{-/-} KCs, *Pkm2*^{-/-} BMDMs, *Gpr3*^{-/-} BMDMs, *Gpr3*^{-/-} ImKCs and *Arrb*^{-/-} ImKCs all had a lower level of basal ECAR; and they did not elevate ECAR in response to 50 nM DPI. Although they responded to 500 nM DPI but the magnitude of increase in ECAR was significantly lower than their wildtype counterparts. The reviewer raised an important question whether *Pkm2*^{-/-} KCs have functional metabolic pathways.

The following observations suggest that *Pkm2*^{-/-} KCs (or *Pkm2*^{-/-} BMDMs, *Gpr3*^{-/-} BMDMs, *Gpr3*^{-/-} ImKCs and *Arrb*^{-/-} ImKCs) have functional metabolic pathways. First, all the knockout cells, including *Pkm2*^{-/-} KCs, responded to glucose, oligomycin and 2-DG although they have a lower basal level of ECAR, suggesting that they are metabolically active, probably because of compensation by PKM1 or PKM2 is not essential for maintaining a minimal level of basal glycolytic activity (See reference 13 and 46). Second, we were able to isolate similar numbers of KCs from KC-specific *Pkm2*^{-/-} mice and control *Pkm2*^{fl/fl} mice and there was no noticeable difference in their survival following culture *in vitro*, suggesting that Pkm2-deficiency does not compromise KC differentiation and survival in mice. Similarly, we did not notice any difference in differentiation, survival and growth between *Pkm2*^{-/-} BMDMs, *Gpr3*^{-/-} BMDMs, *Gpr3*^{-/-} ImKCs and *Arrb*^{-/-} ImKCs and their wildtype counterparts. Finally, we did not observe any growth defect under normal diet between KC-specific *Pkm2*^{-/-} mice and *Pkm2*^{fl/fl} mice, or even between germline *Pkm2*^{-/-} mice and *Pkm2*^{+/+} controls (see reference 46). These observations support PKM2-deficient KCs have functional metabolic pathways.

We have revised the manuscript to include these results and discuss how PKM2 may shape KC function.

REVIEWERS' COMMENTS

Reviewer #5 (Remarks to the Author):

All previously raised experimental concerns have been sufficiently addressed by the authors. I applaud the authors on a well-executed study that holds promise to move the field forward. One minor suggestion, that would substantially improve the manuscript, is to add to the discussion section a brief paragraph focused on the glycolytic skewing-driven divergence in functional inflammatory outcomes between KCs and other immune cells.

Reviewer's comments are *in italics* and the responses are in regular blue font.

Reviewer #5 (Remarks to the Author):

All previously raised experimental concerns have been sufficiently addressed by the authors. I applaud the authors on a well-executed study that holds promise to move the field forward. One minor suggestion, that would substantially improve the manuscript, is to add to the discussion section a brief paragraph focused on the glycolytic skewing-driven divergence in functional inflammatory outcomes between KCs and other immune cells.

Response: We are glad that the reviewer #5 mentioned our additional studies sufficiently addressed all his/her concerns. To address the minor suggestion, we added more discussions about the glycolysis and inflammation in the discussion section in our revised manuscript.